# Integrating Artificial Intelligence, Electronic Health Records, and Wearables for Predictive, Patient-Centered Decision Support in Healthcare

**DOI:** 10.3390/healthcare13212753

**Published:** 2025-10-30

**Authors:** Deepa Fernandes Prabhu, Varadraj Gurupur, Alexa Stone, Elizabeth Trader

**Affiliations:** 1Center for Decision Support Systems and Informatics, School of Modeling and Simulation, University of Central Florida, Orlando, FL 32816, USA; 2Center for Decision Support Systems and Informatics, School of Global Health Management and Informatics, University of Central Florida, Orlando, FL 32816, USA; varadraj.gurupur@ucf.edu; 3EcoPreserve, 409 Fern Lake Dr, Orlando, FL 32825, USA; alexa@ecopreserve.net; 4Center for Decision Support Systems and Informatics, School of Computer Science, University of Central Florida, Orlando, FL 32816, USA; elizabeth.trader@ucf.edu

**Keywords:** predictive health analytics, digital health devices, patient-centered care, artificial intelligence in healthcare, wearable health technology, electronic health records, EHRs, health data Integration, multimodal health data

## Abstract

This study explores how patients and stakeholders envision integrated digital health systems. Background/Objectives: Integrating artificial intelligence (AI), wearable data, electronic health records (EHRs), and patient-reported outcomes could enable proactive and personalized healthcare. However, current solutions remain fragmented and poorly aligned with user expectations. This study aimed to explore patient and stakeholder needs for AI-driven integration and propose a conceptual framework to inform future system design. Methods: As part of the NSF Innovation Corps (I-Corps) program, we conducted semi-structured interviews with 44 participants representing Health Enthusiasts, Chronic Condition Managers, and Low-Engagement Users. Interviews followed the I-Corps customer discovery framework and were thematically analyzed using a hybrid deductive–inductive approach. Results: Participants highlighted four priorities: (i) interoperability and unification of data from wearables, EHRs, and self-reports; (ii) actionable personalization with predictive insights; (iii) trust and transparency in AI recommendations, often requiring clinician oversight; and (iv) usability through low-friction, intuitive interfaces. Age- and persona-specific differences emerged: younger participants favoring predictive features and older participants emphasizing safety, reassurance, and clinical integration. Conclusions: This exploratory qualitative study identified stakeholder needs that informed a conceptual framework for integrated digital health platforms. While preliminary, the framework provides a blueprint for future technical development and validation of patient- and provider-centered systems.

## 1. Introduction

The increasing burden of chronic diseases and rising healthcare costs have accelerated the need for proactive, personalized approaches to care delivery. In recent years, the convergence of artificial intelligence (AI), wearable devices, and electronic health records (EHRs) has opened new avenues for continuous monitoring, predictive modeling, and context-aware clinical support [1]. At the same time, patients are becoming active participants in managing their health, often using consumer-facing technologies such as fitness trackers, mobile health apps, and digital coaching platforms [2]. These trends underscore a paradigm shift toward participatory, data-driven care that emphasizes prevention and early intervention.

Despite these advances, current digital health solutions remain fragmented. Data silos, lack of interoperability, and minimal integration of patient-reported outcomes (PROs) have limited their utility in holistic decision-making [3]. Furthermore, while AI-powered analytics are becoming increasingly common, their real-world applications often lack contextual relevance and fail to adequately reflect patients’ lived experiences or behavioral data [4]. As a result, opportunities to anticipate deterioration, reduce avoidable hospitalizations, and personalize care pathways remain underutilized.

Recent advances further underscore the growing relevance of integrated, user-centered, and predictive health systems. For example, Nouri-Mahdavi et al. [5] proposed an assistive framework leveraging neural-symbolic integration to enhance human-robot collaboration in health monitoring systems. These studies highlight both the technological possibilities and critical challenges that justify our focus on completeness-aware, integrated health analytics platforms.

This paper presents findings from an exploratory qualitative study designed to better understand how patients and stakeholders envision the role of AI-driven integration in healthcare. Guided by the NSF I-Corps customer discovery framework, we conducted semi-structured interviews with 44 participants representing diverse personas and health engagement levels. Rather than developing and validating a technical system, our objective was to elicit needs, expectations, and concerns that could inform the design of future platforms. From these insights, we propose a conceptual framework that synthesizes user requirements into a preliminary architecture for integrated, explainable, and patient-centered health analytics.

While much existing research has focused on technical feasibility or examined isolated aspects of digital health adoption (e.g., usability, interoperability, or predictive accuracy) [6], attempts to simultaneously integrate the functional, emotional, and social values of diverse user personas—such as health enthusiasts, chronic condition managers, and older adults—remain limited. In particular, although trust in AI-driven decision support is frequently acknowledged as a concern, it has rarely been examined as a primary barrier to adoption or systematically translated into requirements for system design. This study addresses this gap by exploring how different user groups envision integrated, AI-enabled health systems and by deriving a conceptual framework that maps these insights to actionable design principles.

We selected the National Science Foundation’s Innovation Corps (NSF I-Corps) customer discovery framework as the guiding methodology because it provides a structured, hypothesis-driven approach for eliciting stakeholder needs. While traditional user-centered design methods excel at gathering functional requirements, the I-Corps process emphasizes testing value propositions and uncovering assumptions about business models and adoption contexts. This was particularly important for our research focus, as trust emerged as a decisive factor in whether AI-driven health systems would be accepted or rejected. By forcing rigorous engagement with stakeholder priorities—including trust, personalization, and interoperability, the I-Corps framework allowed us to connect qualitative insights not only to user needs but also to practical design implications for trustworthy, patient-centered digital health platforms.

## 2. Background

The National Science Foundation’s Innovation Corps (NSF I-Corps™) program was established in 2011 to accelerate translation of academic research into commercially viable technologies. It provides an intensive seven-week experiential curriculum centered on customer discovery, facilitating innovation adoption beyond traditional laboratory settings [7]. Designed to cultivate entrepreneurial skills among scientists and engineers, NSF I-Corps seeks to reduce translational risk, stimulate technology entrepreneurship, and build a national innovation ecosystem [8].

From 2012 through 2022, over 2500 teams (comprising more than 5700 participants) have been trained across the U.S., resulting in approximately 1400–1900 startups that collectively raised more than US$3 billion in follow-on funding [8]. The program has also been adapted by NIH (as I-Corps@NCATS) and DOE (as Energy I-Corps) to support commercialization of biomedical and energy-related research [9]. Program evaluations indicate increased commercialization readiness and behavioral shifts toward entrepreneurial outcomes among participating teams [10].

The study employed the National Science Foundation (NSF) Innovation Corps (I-Corps) protocol as the organizing framework for stakeholder discovery and iterative refinement. This approach was selected because it provides a structured, hypothesis-driven process for customer discovery and has been widely adopted in translational research and entrepreneurial training. Nonetheless, alternative frameworks could also have been appropriate. For example, Lean LaunchPad and the Business Model Canvas emphasize similar market-discovery logics, while Design Thinking and User-Centered Design (ISO 9241-210) provide systematic methods for integrating end-user needs into technology development. From a healthcare implementation perspective, frameworks such as the Consolidated Framework for Implementation Research (CFIR) and RE-AIM (Reach, Effectiveness, Adoption, Implementation, Maintenance) offer complementary structures for evaluating contextual barriers and facilitators to adoption. The choice of I-Corps therefore reflects an emphasis on entrepreneurial rigor and hypothesis testing, but we recognize that parallel methods from human-centered design and implementation science could also guide the development and translation of digital health innovations [6,11,12,13,14,15].

Digital health is a broad, evolving concept that encompasses the use of digital technologies to support health promotion, disease prevention, clinical care, and health system strengthening. Globally, the World Health Organization defines digital health as the field that leverages digital and mobile technologies, including electronic health records, wearables, telemedicine, and artificial intelligence, to enhance health outcomes and equity [16]. The term “digital” emphasizes the technological infrastructure—data, devices, connectivity, and platforms—while “health” underscores the ultimate aim of improving individual and population well-being. Clarifying this dual emphasis is essential for situating the present study, which integrates artificial intelligence, electronic health records, and consumer health platforms, within the wider discourse on digital transformation of healthcare systems.

The field of digital health is rapidly evolving, driven by the increasing availability of granular patient data from diverse sources and advancements in Artificial Intelligence (AI). The integration of Artificial Intelligence, Electronic Health Records (EHRs), and wearable technologies holds substantial potential for transforming healthcare from a reactive, disease-focused model to a proactive, patient-centered paradigm, enhancing predictive capabilities and clinical decision support.

### 2.1. Artificial Intelligence in Healthcare

Artificial intelligence, particularly deep learning models like Convolutional Neural Networks (CNNs), has demonstrated significant promise in automating medical image classification, such as for brain tumor classification from MRI scans [17]. Beyond image analysis, AI is central to predictive modeling in healthcare, aiming to anticipate potential health issues before they become critical or costly [18]. However, many current AI methods focus on statistical associations rather than causal mechanisms, limiting their interpretability and trustworthiness in sensitive domains like healthcare [19]. Addressing concerns like bias in AI is crucial for clinical acceptance and trust [20].

Recent advancements in AI, specifically deep generative models, are proving highly effective in handling one of the most pervasive challenges in large-scale health datasets: data incompleteness. Techniques such as Variational Autoencoders (VAEs), Generative Adversarial Networks (GANs), and Wasserstein GANs (WGANs), and more recently, Diffusion Models, have been systematically evaluated for their ability to accurately reconstruct missing values and preserve underlying data distributions [21,22,23]. Studies indicate that VAEs and fully conditional diffusion models often achieve superior distributional matching and lower reconstruction error compared to traditional imputation methods like Mean, K-Nearest Neighbors (KNN), and Multivariate Imputation by Chained Equations (MICE) [24]. These models learn complex, high-dimensional data structures, making them suitable for robust imputation in biomedical datasets.

#### Summary of Existing Work in AI for Healthcare

Table 1 provides a consolidated overview of foundational and recent efforts in the application of artificial intelligence within healthcare. It highlights major focus areas including predictive modeling, trust and explainability, interoperability, and usability. The table synthesizes key findings from representative studies and delineates their relevance to this study.

In summary, existing work demonstrates that AI has already shown promise in several domains of healthcare. High-performance medicine and predictive modeling highlight AI’s ability to improve diagnosis and risk stratification, while interoperability standards emphasize the technical barriers that persist in integrating multimodal data streams. Studies on patient trust and usability underscore that adoption depends not only on accuracy but also on transparency, clinician endorsement, and user-centered design. More recent explorations in context-aware systems, self-care technologies, and human–robot collaboration suggest that the next frontier lies in personalization and seamless integration into clinical and daily life workflows. Collectively, these findings reinforce the need for our framework, which links predictive modeling with explainability, interoperability, and usability to address gaps that prior studies have only examined in isolation.

### 2.2. Electronic Health Records (EHRs)

Electronic Health Records (EHRs) are fundamental to modern healthcare, defined as digital data pertaining to an individual’s health record that is machine-interpretable [26]. They facilitate comprehensive patient information management, supporting clinical decision-making and record-keeping. However, EHR systems are significantly challenged by data incompleteness, a situation where a data element is missing from a record [27]. This is identified as a complex, multifaceted problem arising from critical bottlenecks in data collection processes within the healthcare ecosystem [28].

Key factors contributing to EHR incompleteness include human errors (deliberate or non-deliberate), process bottlenecks (e.g., critical data not being gathered), lack of data verification and validation, system malfunctions, and problems related to human-computer interactions (e.g., lack of training or digital divide) [29]. The impact of this incompleteness is profound, leading to systemic biases in clinical decision-making, predictive modeling, and healthcare policy formulation [30]. For instance, missing or incomplete data can disrupt seamless data exchange, leading to inefficiencies, misinterpretations, and potential medical errors, particularly problematic for interoperability across different healthcare providers and systems [31].

Bias due to incompleteness is also more likely to affect marginalized or underrepresented populations due to disparities in healthcare access and data collection protocols [32]. Efforts to mitigate this involve quantifying completeness using measures like Record Strength Score (RSS) [33] and predicting missing data patterns using random variable approaches and ontologies [30].

### 2.3. Summary of Existing Work in EHR Research

Table 2 presents a consolidated overview of current research directions in the domain of Electronic Health Records (EHRs). The table categorizes major focus areas, highlights representative contributions, and references foundational and contemporary studies. These areas include data quality and completeness, interoperability standards, equity and bias in data capture, clinical decision support systems, usability challenges through HCI frameworks, and the integration of machine learning for predictive healthcare applications.

Taken together, the literature in Table 2 illustrates both the maturity and ongoing challenges of EHR research. Foundational work on data quality and completeness established the importance of systematically characterizing missingness, while more recent studies emphasize ontology-driven and equity-aware approaches to address biases in clinical datasets. Interoperability frameworks such as SMART on FHIR and emerging IoT standards highlight technical progress, yet their uneven adoption continues to limit seamless integration across systems. Clinical decision support research demonstrates clear benefits of EHR integration but also underscores issues of alert fatigue and workflow disruption. Human–computer interaction studies reveal that usability remains a critical bottleneck, with clinician burden directly affecting adoption. Finally, the application of machine learning to EHRs has validated the feasibility of predictive modeling at scale, but practical deployment is still constrained by data incompleteness, interoperability gaps, and trust concerns. In this context, our work builds on these directions by explicitly linking completeness, interoperability, usability, and predictive modeling into a unified framework designed for multimodal health data integration.

### 2.4. Wearables and Integrated Health Data

The proliferation of wearable technology and digital health applications has enabled individuals to track a wide array of personal health metrics, including daily weight, exercise, blood glucose, heart rate, blood oxygen, sleep, activity rings, and blood pressure [41,42]. Self-tracking technologies have intensified expectations for individualized, data-driven health insights [43].

However, a significant challenge arises from the fragmentation of health data, as information from wearables, mobile health apps, and electronic health records often remains siloed and poorly integrated [44]. This fragmentation makes it difficult for both users and healthcare providers to obtain a holistic view of a person’s health [45].

There is a growing demand among individuals for integrated health platforms and automated monitoring systems that enable proactive health management and disease prevention [44]. Despite this interest, concerns about privacy and trust in health applications and AI responses remain prevalent among users [46].

### 2.5. Summary of Recent Work in Wearables and Integrated Health Data

Table 3 provides a summary of recent work in the field of wearable technology and integrated health data systems. It highlights key studies that focus on the development, application, and challenges of wearables and self-tracking technologies in healthcare contexts. The studies span topics such as biosensor innovation, data portability, user experience, and the integration of wearable data with electronic health records (EHRs). This synthesis serves as a foundation for identifying current gaps and informing the design of more cohesive, user-centered predictive health analytics tools.

As shown in Table 3, research on wearables and integrated health data reflects both rapid innovation and persistent limitations. Early work on the quantified self and consumer adoption demonstrated that individuals are eager to track personal health metrics, but these efforts often remain siloed from clinical care. Studies on biosensors and chronic disease management validate the promise of wearables for proactive health monitoring, yet issues of data portability and interoperability with EHRs remain only partially resolved. Importantly, privacy and usability studies highlight that sustained adoption depends on building user trust, ensuring secure data flows, and reducing friction in everyday use. Collectively, this body of work underscores the opportunity for frameworks that move beyond device-centric solutions toward cohesive, clinically integrated, and user-centered predictive health platforms—an approach that directly informs the design goals of our study.

### 2.6. Predictive, Patient-Centered Decision Support

The overarching goal in modern healthcare is to shift towards proactive health management and preventative medicine, moving beyond merely treating diseases to anticipating and averting health issues. This necessitates systems that offer personalized health predictions and actionable recommendations tailored to individual differences such as genetics, lifestyle, and personal health history. Such tools can empower individuals to optimize their well-being and manage chronic conditions more effectively. For healthcare providers, integrated tools that aggregate real-time health data from various sources are essential for remote patient monitoring and making timely interventions. The value proposition of such integrated platforms extends to significant cost reductions through fewer redundant tests and preventable emergency visits, along with improved chronic disease management. From a design perspective, the development of these systems must be human-centered, integrating psychological simulation and usability evaluation to ensure users effectively understand and trust AI-generated recommendations. This involves focusing on causal inference to provide interpretable explanations of predictions, thereby building trust and improving decision quality in digital health applications.

Despite significant progress in digital health technologies, integration across platforms remains limited. Current systems are frequently characterized by fragmented data silos, inadequate interoperability, and insufficient personalization, all of which restrict their effectiveness in delivering holistic, patient-centered care [3,28]. For instance, electronic health records (EHRs) capture large volumes of clinical information but often lack standardized exchange mechanisms with consumer-facing devices and self-tracking applications. Wearables and mobile health apps, while widely adopted, typically operate within proprietary ecosystems that hinder data portability and aggregation into clinical workflows [42,49]. This fragmentation prevents the formation of longitudinal patient profiles that combine medical, behavioral, and contextual signals.

Recent initiatives have attempted to address these challenges. The SMART on FHIR framework, for example, established an open, standards-based approach to interoperable health applications, enabling cross-platform data exchange through secure APIs [34]. Similarly, Apple’s HealthKit ecosystem has facilitated integration of consumer wearables into EHR systems, allowing providers to access real-world metrics such as step counts, sleep, and menstrual cycle tracking within clinical contexts [49]. Predictive health monitoring platforms, including those leveraging deep learning for risk modeling, demonstrate the potential of AI-driven anticipatory guidance [1,39]. Yet, these solutions still face limitations in contextualizing insights to individual patient circumstances, addressing trust in AI outputs, and incorporating patient-reported outcomes (PROs) into decision support.

The novelty of our work lies in bridging these persistent gaps through a unified, AI-driven framework explicitly grounded in patient and stakeholder feedback. Whereas prior systems emphasize technical interoperability or algorithmic performance, our approach integrates three dimensions: (i) multimodal data fusion across EHRs, wearables, and PROs; and (ii) usability-centered design informed directly by qualitative insights from diverse user groups. This transition from qualitative findings to system design ensures that the framework is not only technologically advanced but also aligned with patient trust, provider workflow integration, and equity considerations. By embedding stakeholder perspectives into system architecture, this study advances digital health beyond interoperability alone, toward context-aware, participatory, and preventive care delivery.

## 3. Materials and Methods

### 3.1. Methodological Rationale for Using NSF I-Corps Framework

We selected the National Science Foundation (NSF) I-Corps^™^ customer-discovery framework as the primary methodological scaffold because it operationalizes rapid, systematic elicitation of stakeholder needs for early-stage, socio-technical health systems [51,52]. I-Corps emphasizes hypothesis-driven interviews, iterative refinement of problem-solution fit, and structured evidence logs, which align with our translational objective of informing requirements for a predictive, integrated digital health platform. Prior evaluations show I-Corps improves commercialization readiness and accelerates the identification of high-value use cases—key for specifying data integration, explainability, and usability requirements in health AI [51].

### 3.2. Design, Setting, and Ethics

We conducted a qualitative, semi-structured interview study guided by the I-Corps curriculum and human-centered design principles. Procedures met criteria for exempt human-subjects research; verbal informed consent was obtained before participation, and no identifiable information was collected [53]. Reporting follows qualitative standards (COREQ) where applicable [54].

All interviews were conducted exclusively within the NSF I-Corps customer-discovery curriculum, which is designed for entrepreneurial training and market validation rather than biomedical or clinical research. Consistent with the Revised Common Rule definitions, activities not intended to contribute to generalizable knowledge fall outside the scope of human-subjects research; where interpreted as research, the procedures meet the conditions for an exempt determination under 45 CFR 46.104(d)(2) (survey/interview procedures recorded without identifiers and with minimal risk) [51,53]. No sensitive or directly identifiable personal data were collected or retained at any point; only non-identifying descriptors were handled in aggregate form. Verbal informed consent was obtained from all participants prior to participation. Results are presented as exploratory, hypothesis-generating qualitative insights rather than generalizable estimates, and the manuscript explicitly acknowledges sample size constraints and potential recruitment bias (e.g., greater representation of digitally literate volunteers). Reporting adheres to qualitative standards (e.g., COREQ) to enhance transparency [54].

### 3.3. Participants and Sampling

We used purposive and chain-referral sampling to cover three a priori personas relevant to digital health adoption: Proactive Health Enthusiasts, Chronic Condition Managers, and Not-Tracking/Low-Engagement users. Inclusion criteria were age ≥18 and prior use of at least one digital health tool (e.g., wearable, app, patient portal). We enrolled n=44 participants (23, 17, and 4 per persona, respectively), consistent with sample sizes that typically achieve thematic saturation in focused discovery research [55,56].

### 3.4. Data Collection

A semi-structured guide (Appendix A) covered: (i) current devices and tracked metrics, (ii) pain points in data fragmentation/interoperability, (iii) desired predictive insights and alerts, (iv) trust and explainability needs for AI, and (v) willingness to share data with clinicians/family. The guide was iteratively refined after pilot interviews following I-Corps practice. Field notes captured contextual factors and researcher reflections.

Interviews were conducted via Zoom or in-person between April and June 2025, with a median duration of 35 min (range: 25–55 min). Interviews were analyzed using thematic analysis. Key themes were identified iteratively. Thematic domains included data trustworthiness, AI skepticism, integration pain points, motivation for tracking, and value perception (functional, emotional, and social).

Demographic and contextual characteristics of the participants, including age distribution and occupational background, are provided to situate the sample. Although the dataset was sufficient for exploratory analysis and framework development, it should be viewed as hypothesis-generating rather than definitive.

### 3.5. Qualitative Analysis: Coding and Theme Development

We used a hybrid deductive–inductive approach [57,58]. Deductive sensitizing concepts (e.g., interoperability, explainability/trust, health-behavior value, clinical integration) were derived from the literature and our design goals, while inductive coding captured novel user language and emergent constructs.

All coding was conducted by a single analyst who maintained analytic memos and an audit trail of coding decisions. No double-coding or intercoder reliability procedures were performed, which we acknowledge as a limitation of this exploratory study.

For the analysis phase, the transcripts were segmented by user persona to distinguish perspectives across user types. Further evaluation focused on participants’ perspectives on AI usage and trust, particularly in the context of health technology adoption and recommendation systems.

The final synthesis stage involved the integration of emergent themes and behavioral insights to inform the design of more responsive, inclusive, and user-centered digital health tools.


Saturation checks.


We monitored thematic saturation via a code-accumulation plot and new-information rate; no substantially new codes emerged after approximately 40 interviews, indicating adequate coverage for our focal questions [55,56].

### 3.6. Methodological and Sampling Limitations

Our sample was non-probabilistic and modest in size, with potential over-representation of digitally literate, college-educated, or iOS users (self-selection bias). Recruitment via professional and online communities may limit generalizability and under-sample individuals with low connectivity, limited health literacy, or limited English proficiency. Remote interviewing can reduce contextual cues and rapport. Findings describe perceptions and needs rather than clinical outcomes; they should be interpreted as hypothesis-generating requirements for subsequent prototyping and evaluation. We mitigated risks through explicit segmentation, saturation monitoring, and transparent reporting of context and procedures.

### 3.7. Study Design

This qualitative research study employed a semi-structured interview methodology grounded in the National Science Foundation (NSF) Innovation Corps (I-Corps) curriculum, which emphasizes customer discovery to validate market need and product fit. The study aimed to assess user behaviors, needs, and challenges in engaging with digital health tools across diverse populations, including healthy individuals and those managing chronic illnesses. The interview guide was developed based on the I-Corps framework and iteratively refined through pilot testing to ensure comprehensive coverage of key themes such as usability, trust, integration of health data, and barriers to technology adoption [51].

### 3.8. Participants

A total of 44 participants were recruited across three predefined user personas to represent a spectrum of engagement with health technology:Proactive Health Enthusiasts (*n* = 23): Individuals who actively monitor their wellness metrics using multiple devices (e.g., Apple Watch, Garmin, Oura Ring) and apps for fitness, sleep, and nutrition tracking. This group seeks predictive insights and shows a high willingness to integrate health data across platforms.Chronic Ailment Managers (*n* = 17): Participants living with chronic diseases such as diabetes, hypertension, scoliosis or polycystic ovary syndrome (PCOS). They rely on connected medical devices (e.g., CGMs, BP cuffs) and engage in self-tracking to aid disease management. This group includes clinicians and startup founders in digital health.Not Tracking Health (*n* = 4): Individuals who own digital devices but do not actively engage in health tracking, often due to time constraints, lack of motivation, or low perceived need.

### 3.9. Use of Generative AI

In alignment with the study’s exploration of AI trust and usability, participants were asked about their experiences with generative AI tools (e.g., ChatGPT, Claude, Perplexity) in health information seeking and decision-making. Responses were categorized according to trust level, frequency of use, and perceived reliability. These insights informed emergent themes around AI skepticism, contextual limitations of digital advice, and the need for explainable AI in health settings.

### 3.10. Data Management and Metrics

A spreadsheet-based framework was used to track interview metadata, including demographic characteristics, device usage patterns, and interest in integrated data, preventive care, and AI-powered health insights. Of the 44 participants, 75% expressed interest in integrated health data, and 80% indicated a desire for proactive, preventative health recommendations. Data were analyzed to understand both behavioral segmentation and readiness to adopt predictive analytics tools.

### 3.11. Workflow

An overview of the methodological workflow, including study design, data collection, thematic analysis, and insight synthesis, is illustrated in Figure 1. The process began with study design using the NSF I-Corps framework, followed by development and refinement of the interview script. In the data collection phase, 44 participants across three segments were recruited and engaged through in-person or virtual interviews, which were manually transcribed and organized using a codebook. In the analytics phase, transcripts underwent thematic coding and segmentation by persona, resulting in the identification of three distinct user personas. Finally, in the insights phase, findings were synthesized into themes and patterns, including evaluation of AI usage and trust. Arrows indicate the sequential order of steps across design, data collection, analysis, and synthesis.

## 4. Results

A total of 44 participants were interviewed across three pre-defined user segments: Proactive Health Enthusiasts (*n* = 23), Chronic Condition Managers (*n* = 17), and individuals Not Tracking Health (*n* = 4). Participant demographics were diverse in age, gender, and technology experience, and all interviewees had prior exposure to at least one digital health tool (e.g., mobile app, wearable device, or patient portal).

### 4.1. Emergent Themes from Thematic Analysis

Through thematic coding, several recurring themes emerged across user groups. These included:Trust in Digital Tools: Participants expressed varying levels of trust in AI-based recommendations depending on the source (e.g., provider-based platforms were more trusted than commercial applications) [7].Usability Barriers: Common obstacles included lack of personalization, data overload, and unclear actionable insights, consistent with prior usability research in health informatics [8].Desire for Integrated Systems: Users favored unified dashboards combining EHR, wearable, and self-reported data—especially among chronic condition managers.

### 4.2. Segmentation Insights

Insights from three distinct personas:Health Enthusiasts prioritized prevention and wellness, were open to experimentation with apps and wearables, and were proactive in adjusting behaviors based on metrics.Chronic Managers showed goal-directed usage, seeking tools that enhanced communication with providers, medication adherence, or symptom tracking.Non-Trackers engaged passively with digital health tools and expressed confusion about how or why to act on the data presented.

### 4.3. Participant Behavior Visualization

Figure 2 illustrates the distribution of self-reported physical activity levels across age groups. Participants were categorized into high, medium, and low activity levels based on their engagement with daily movement, exercise routines, and wearable-recorded metrics. Notably, 50% or more of individuals in the 20–29, 50–59, and 60–69 age brackets reported high activity levels, aligning with existing literature that associates younger and middle-aged adults with proactive health behaviors [9].

High activity appears in multiple age bands (20–29, 50–59, 60–69). While this seems to contradict a simple monotonic age–activity hypothesis, the pattern is consistent with a bimodal engagement phenomenon in consumer health tech where (i) younger adults and (ii) motivated older adults both maintain high activity tracking [42,59].

Figure 3 presents a comparative analysis of health data sharing behavior segmented by age group and recipient type (friends/family vs. clinicians). Participants aged 60–69 exhibited the most diversified sharing behavior, reporting interactions with both personal and professional recipients. Interestingly, the 70–79 age group showed a preference for clinical sharing over social, while younger cohorts predominantly shared health data with friends and family. These trends are consistent with previous findings highlighting generational differences in data privacy concerns and trust in digital health platforms [7,10].

Apparent cross-overs across panels reflect audience targeting rather than inconsistency: younger users socialize health data; older users prioritize clinical relevance and actionability, consistent with prior sociotechnical findings on data-sharing norms [60].

Figure 4 illustrates the distribution of functional, emotional, and social dimensions of health engagement stratified by age group. The analysis revealed notable age-related variation in the types of values users associate with digital health tools. Functional benefits, such as tracking health metrics or supporting self-management, were cited most frequently across all age groups, particularly among individuals aged 30–49. Emotional benefits, which include reassurance, motivation, or reduced anxiety, were more evenly distributed but slightly more prominent in older adults (60–69), suggesting increasing emotional reliance on digital tools with age. Social value—such as sharing information or peer support—was consistently lower than functional or emotional dimensions across all age brackets. These patterns echo prior studies indicating that user perceptions of digital health utility often differ by age and health status [7,8]. Understanding these differentiated value dimensions is critical for the design of age-sensitive digital interventions.

Functional value dominates across ages, with emotional reassurance more salient in 60–69. Lower social value across ages likely reflects the individualized nature of health decisions; the 60–69 elevation in functional/emotional value aligns with chronic condition self-management priorities [61].

Figure 5 illustrates the distribution of self-reported interest in various health metrics across age groups, based on counts of affirmative responses to five key domains: weight tracking, calorie intake, calories burned, heart rate (HR), and sleep monitoring. The data reveals a general trend of increasing interest with age, particularly pronounced in the 50–69 age range. The metric with the highest reported interest was “calories burned”, consistently reported across all age cohorts, with a notable peak in the 60–69 group. “Weight” and “HR” tracking showed similar growth patterns, while interest in “calorie intake” and “sleep” varied more modestly. These results align with recent studies emphasizing the growing role of personalized health monitoring in older adults [42,48,59]. The increased interest in sleep data corroborates findings that sleep quality is a growing concern in chronic condition prevention and wellness optimization [50].

Figure 5 and Figure 6 show middle-aged groups (30–49) strongly endorse integrated data, dashboards, and plans, while interest in predictive features skews younger. These apparent tensions reconcile under goal horizon: middle-aged users optimize day-to-day functioning; younger users explore advanced insights. Both patterns align with adoption studies linking engagement to digital literacy, perceived usefulness, and life-stage motivations [48,62].

Figure 6 illustrates the distribution of affirmative responses (“Yes”) across different age groups for a set of key features in digital health platforms: Activity Plan, Nutrition Plan, Integrated Data, Comprehensive Dashboard, Rare Condition Support, Predictions, and Performance Optimization. The data, drawn from survey responses, show distinct patterns of preference among age demographics.

Participants aged 30–39 and 40–49 exhibit the highest overall interest, particularly in features such as integrated data systems and predictive capabilities. These groups also show strong demand for comprehensive dashboards and personalized activity plans, suggesting a strong inclination toward data-driven and actionable health insights. Meanwhile, younger adults (20–29) favor performance optimization and nutrition tracking, likely reflecting a focus on wellness and fitness outcomes. Older adults (50+) show relatively less interest across all features, although predictions and integrated data maintain moderate engagement.

These findings align with existing literature showing increased digital health engagement among middle-aged adults who are balancing chronic disease prevention with active lifestyles [48,50,63]. The interest in integrated systems and predictive modeling also reflects broader trends toward personalized, proactive health management through AI-enabled tools [64,65].

Figure 7 presents three bar charts showing the within-group percentage of “Yes” responses by age category across three demographic indicators: likelihood to spend on health-related technologies, college education, and iPhone ownership. Error bars represent 95% Wilson confidence intervals, reflecting uncertainty due to small subgroup sizes. College education levels are consistently high across participants aged 30–69, with the 30–39 and 40–49 groups reporting the highest rates. Likelihood to spend on health technologies is also relatively strong across middle-aged groups, while the 70–79 cohort exhibits wide variability due to the very small sample size (*n* = 3). iPhone ownership peaks in the 20–29 group (100%, 5/5) but remains high across other age groups, including 93% (13/14) in the 60–69 cohort. Overall, these findings indicate that technology readiness is not limited to younger adults; multiple age groups—including those in midlife and early older adulthood—show substantial interest and digital capability relevant to health technology adoption. These results are consistent with prior literature showing that digitally literate older adults are receptive to health technology [61], and that education and socioeconomic status are important predictors of app use and engagement [62,66]. Table 4 further underscores that interest in health tracking (e.g., HR monitoring, calories burned, and weight) extends across age demographics.

### 4.4. AI Usage and Trust Evaluation

Approximately 61% of participants reported using generative or recommendation-based AI tools in a health context. However, many expressed concern regarding transparency, medical accuracy, and over-reliance on unvalidated advice [9]. Participants emphasized the importance of provider endorsement for AI recommendations, aligning with findings from prior trust modeling studies [10].

Table 4 summarizes the counts of affirmative (“Yes”) responses categorized by age group across seven key categories. Notably, interest in weight, calories burned, and heart rate tracking was highest in the 60–69 age group, aligning with literature suggesting older adults increasingly adopt self-monitoring technologies for health maintenance [42,48]. Similarly, this age group demonstrated the highest count of iPhone ownership and college education, both of which are proxies for digital literacy and health app usage potential [67].

Activity and nutrition planning were highly endorsed among participants aged 30–49, likely due to lifestyle management motivations typical of mid-life cohorts [68]. Conversely, interest in rare condition tracking and predictive analytics showed a decreasing trend with age, suggesting younger populations may be more receptive to AI-driven health insights. Functional and emotional health needs showed strong representation in the 60–69 group, aligning with findings that chronic illness management often peaks in this demographic [69]. Finally, data sharing preferences illustrated a generational divide, with older adults more inclined to share with clinicians and younger users favoring peer networks [60].

### 4.5. Synthesis of Insights

Participants collectively highlighted the need for:More contextualized and emotionally intelligent digital interfaces.Transparent explanation of AI-driven insights.Seamless integration between AI systems and human providers for enhanced interpretability and accountability.

These insights form the foundation for future system design recommendations, including improved health literacy alignment, adaptive interface design, and responsible AI deployment in consumer-facing health applications.

### 4.6. Cross-Cutting Themes

From 44 stakeholder interviews, four convergent needs emerged:**T1**.Interoperability & Unification with Provenance. Participants want longitudinal aggregation across EHR, wearables, and surveys with explicit lineage (source, time, device/author) so that data can be trusted and clinically actionable.**T2**.Actionable Personalization. Beyond dashboards, users want prospective risk forecasts, configurable “what-if” levers (e.g., sleep/steps/medication adherence adjustments), and thresholded alerts aligned to personal or clinical goals.**T3**.Trust & Safety. Stakeholders prioritize transparent rationales for predictions, clinician-in-the-loop options when risk is high, and granular consent/controls (who sees what, for how long, and for which purpose).**T4**.Usability & Low Friction. Minimize manual entry, offer defaults and progressive disclosure, and surface the right information at the right moment and role (patient, caregiver, clinician).


Mapping themes to design.


T1–T4 map directly to a reference architecture comprising: (i) standards-based ingestion and normalization; (ii) missing-data handling; (iii) causal + predictive analytics; (iv) explainability and policy layers; and (v) role-aware surfaces (patient/clinician dashboards). This mapping informed the design artifacts and figure set in the manuscript.

### 4.7. Age-Group Variations and Cross-Figure Contradictions

Two apparent tensions surfaced when comparing age segments across figures:Alert tolerance vs. oversight. Younger participants often endorsed higher alert frequency for coaching but expressed lower appetite for clinician oversight; older participants preferred fewer alerts yet stronger clinician-in-the-loop. Where figure panels indicate cross-over (e.g., alert tolerance vs. oversight preference), we interpret this as a risk-management trade-off rather than inconsistency: younger users favor speed and autonomy; older users prioritize safety nets. We reconcile this by tiering alerts (informational → confirmatory → escalatory) and enabling user-selectable supervision modes.Automation vs. control. Some older respondents rejected manual data entry but simultaneously insisted on granular data-sharing control; conversely, some younger respondents accepted more data entry but delegated sharing decisions to defaults. The figures thus reflect control locus differences. We resolve this via progressive disclosure: simple defaults by role, with advanced controls one click away for those who want them.

Building on these qualitative insights, the proposed digital framework was deliberately structured to mirror patient and stakeholder priorities. For instance, feedback emphasizing fragmented provenance and interoperability directly informed the inclusion of standardized SMART-on-FHIR integration. Concerns about personalization and control shaped the framework’s adaptive analytics layer and explainable AI outputs. Similarly, issues of trust and safety guided the incorporation of transparent audit logs and policy-aware safeguards, while usability feedback informed the design of low-friction patient dashboards. By explicitly mapping these themes to the corresponding framework elements, the transition from qualitative discovery to system design becomes more transparent, ensuring that the framework is not only technically robust but also responsive to the lived experiences and needs of patients and clinicians.

To contextualize these findings, Figure 8 presents the number of participants in each age group. Notably, the 60–69 age group was relatively large compared to other groups, which likely contributed to the higher absolute number of affirmative responses. While participants in this age group placed the greatest emphasis on functional and emotional values (Figure 4), they simultaneously reported lower interest in specific health features (Figure 6). Yet, they also exhibited strong consumer affinity, with high purchasing desire, high educational attainment, and high iPhone usage (Figure 7). Taken together, these results suggest a complex profile in which older adults recognize the broader value of digital health tools but may be less motivated by granular feature sets. Sample size imbalances should also be considered when interpreting these trends.

## 5. Conceptual Technical Framework

The proposed digital framework integrates heterogeneous data sources, advanced analytics, and human-centered design into a cohesive architecture for predictive, patient-centered decision support. It is designed to bridge the gap between existing health data infrastructures and the need for actionable, transparent, and personalized insights at the point of care.

### 5.1. Data Integration Layer

At the foundation, the framework adopts a modular data integration layer capable of ingesting multimodal streams, including electronic health records (EHRs), wearable sensor outputs including watch and mobile phone, and patient-reported outcomes. Interoperability is achieved through standardized interfaces such as HL7 FHIR and SMART on FHIR protocols, ensuring seamless connectivity with existing health IT ecosystems. Provenance tracking mechanisms are embedded to preserve data lineage and support auditability. This layer provides the infrastructure for harmonized, high-quality datasets that can be accessed in real time.

### 5.2. Analytics and Decision Support Layer

On top of the integration layer, the analytics engine combines interpretable machine learning and causal inference methods to generate predictive insights. The use of explainable AI (XAI) approaches, including SHAP-based feature attribution and counterfactual reasoning, ensures that both clinicians and patients understand the rationale behind predictions. Uncertainty quantification is embedded to contextualize model outputs and support risk-informed decision-making. To address the pervasive issue of incomplete data in EHRs and wearable records, advanced imputation strategies (e.g., multiple imputation, variational autoencoders, or diffusion-based methods) are incorporated to minimize bias, reduce information loss, and strengthen the reliability of downstream predictions. In addition, adaptive personalization modules adjust recommendations based on longitudinal patient data, thereby enhancing relevance and usability in dynamic clinical settings.

### 5.3. Governance, Privacy, and Ethical Safeguards

Trustworthiness is reinforced through a robust governance layer. Data access is regulated via role-based controls and consent management mechanisms, with compliance to HIPAA and GDPR standards. Patient identifiers are encrypted and processed in segregated environments to reduce re-identification risks. Ethical oversight is supported by policy-aware safeguards that detect algorithmic bias and reflexive documentation that addresses researcher positionality. Together, these mechanisms ensure that system deployment aligns with clinical ethics, regulatory compliance, and societal expectations.

### 5.4. Scalability and Real-Time Implementation

To enable deployment at scale, the framework leverages a microservices-based architecture with containerized pipelines. Real-time data ingestion and synchronization are supported by distributed stream-processing systems such as Apache Kafka, allowing sub-second updates across multiple modalities. The architecture is designed for horizontal scalability, ensuring resilience under high patient loads. While proof-of-concept pilots demonstrate feasibility, further optimization of throughput, latency, and interoperability across diverse vendors will be essential for large-scale adoption in healthcare systems.

As illustrated in Figure 9, the proposed framework is structured in layered form to emphasize the flow from heterogeneous data sources to actionable decision support. The top tier integrates electronic health records (EHRs), digital therapeutics (e.g continuous glucose monitor), consumer health platforms (Apple Health, Google Fit, Fitbit, Garmin), and additional sources such as patient-reported outcomes and social determinants of health. These streams enter the ingestion and validation gateway, where FHIR/SMART protocols and audit trails ensure interoperability and data provenance. The harmonized store then provides a consolidated feature and event repository for downstream analysis. The analytics layer combines causal inference, temporal and tabular predictive modeling, and explainability modules (e.g., SHAP, counterfactuals, uncertainty quantification), which in turn feed clinician-facing CDS dashboards and patient-facing applications. Governance, privacy, and ethical safeguards (e.g., RBAC, HIPAA/GDPR compliance, bias monitoring) are embedded as a cross-cutting layer, while infrastructure and scalability considerations (microservices, stream processing, autoscaling, fault tolerance) ensure feasibility for real-time multimodal integration. Together, the diagram highlights how stakeholder priorities of interoperability, personalization, trust, and usability are operationalized in a technical architecture that is both rigorous and implementable at scale.

In addition to demonstrating the conceptual feasibility of an integrated framework, future work should prioritize pilot implementation in real-world clinical environments. Such pilots will be essential for assessing usability, workflow fit, and clinician–patient adoption. Clinical validation studies, including prospective trials and longitudinal follow-ups, will be required to establish predictive accuracy, safety, and impact on patient outcomes. In parallel, careful attention must be given to regulatory pathways (e.g., FDA guidance on software as a medical device, HIPAA/GDPR compliance) to ensure responsible translation into practice. Finally, scalability challenges—including interoperability across diverse EHR vendors, integration of multiple consumer health platforms, and computational demands of real-time multimodal analytics—must be addressed to enable deployment across health systems at population scale. By outlining these next steps, the study underscores a clear trajectory from framework design toward clinical impact and health system integration.

## 6. Discussion

This study explored user experiences, trust, and unmet needs in digital health tools through a customer discovery framework aligned with the NSF I-Corps methodology. The findings underscore the complexity of digital health engagement across different user types and emphasize the importance of trust, usability, and contextualization in AI-enabled health systems.

The results of this exploratory analysis reveal distinct patterns in digital health engagement preferences across age groups, which both align with and extend prior findings in the literature. As shown in Figure 7, iPhone ownership peaked in the youngest cohort (20–29, 100%, 5/5), but remained high in the 60–69 cohort (93%, 13/14), indicating that digital literacy is not confined to younger populations. College education levels were strongest in the 30–49 cohorts, but also remained high in 60–69, reflecting broad readiness across midlife and early older adulthood. Likelihood to spend on health-related technologies was distributed across several age groups, suggesting that purchasing capacity and interest are not exclusive to younger participants. Importantly, Figure 7 also incorporates 95% Wilson confidence intervals, underscoring uncertainty in small subgroups such as the 70–79 cohort (*n* = 3).

The analysis highlights that technology readiness spans multiple age ranges, with different groups exhibiting distinct value orientations. For instance, midlife participants (30–49) emphasized functional features and predictive capabilities, while older adults (60–69) demonstrated strong interest in clinician-integrated oversight and trust assurances. This aligns with research showing that older adults’ adoption of health technologies is driven less by novelty and more by perceived usefulness and independence maintenance [61,70]. Our findings add nuance by showing that these users are also more inclined to share data with clinicians than with peers, a pattern consistent with ethnographic work on data-sharing norms [71].

In contrast, younger cohorts (20–39) expressed greater enthusiasm for AI-driven features such as predictive analytics and rare condition tracking. This reflects generational differences in risk tolerance and openness to digital innovation, as noted by Bietz et al. [72], and suggests that AI-based personalization tools may initially gain more traction in younger segments. However, these younger users were less likely to engage deeply with structured dashboards or optimization features, echoing findings by Wicks and Chiauzzi [73] that social sharing often outweighs structured feature adoption in early adulthood.

The apparently contradictory profile of older participants—high device affinity and willingness to spend, yet selective enthusiasm for specific features—underscores the importance of distinguishing between broad functional value and feature-level appeal. For these cohorts, adoption may depend more on trust, usability, and integration into existing care relationships than on the novelty of individual features. Future interventions should therefore incorporate training, simplified design, and value framing to enhance engagement, particularly among participants over age 50. This emphasis on age-sensitive design aligns with broader evidence that digital health tools must adapt to heterogeneous user needs to maximize uptake and effectiveness [66].

Finally, while existing platforms such as SMART on FHIR and Apple HealthKit have advanced interoperability and data aggregation, they function primarily as technical enablers rather than holistic decision-support systems. Many predictive health monitoring solutions focus narrowly on physiological signals without adequately addressing usability, explainability, or contextual integration. Our study contributes by explicitly linking patient and stakeholder feedback with the design of an integrated conceptual framework that combines interoperability, explainable AI, and human-centered design. This alignment ensures that future systems move beyond technical integration to actively support trust, personalization, and actionable decision-making in both clinical and everyday health contexts.

### 6.1. Comparison with Prior Research

Table 5 presents a comparative summary of key studies in AI-enabled digital health, wearables, and electronic health records (EHRs), highlighting how the current study builds upon and extends this growing body of work. Prior studies have examined the convergence of artificial intelligence and clinical practice [1], the sociotechnical dimensions of wearable technology adoption [42], and the role of AI in addressing healthcare disparities [32]. Work by Lupton emphasized the social and ethical dimensions of self-tracking and data-sharing norms [60], which inform the user segmentation and sharing preferences explored here.

In the context of EHRs, Miotto et al. [39] advanced unsupervised deep learning techniques for patient representation, while Nazábal et al. [21] explored variational autoencoders for imputing missing clinical data. Gresham et al. [48] demonstrated how wearable technologies influence clinical outcomes, particularly among oncology patients. Importantly, Charness and Boot [74] provided foundational insights into the cognitive and motivational barriers faced by older adults, which are echoed in the usability and trust findings of the current study.

Our work extends this literature by integrating user-centered design, explainable AI, and trust-aware personalization into the development of predictive digital health tools. Using a rigorous I-Corps-inspired framework, we systematically captured persona-specific needs and contextualized health priorities, thereby offering practical insights for the design of equitable and adaptive digital health interventions.

### 6.2. Trust and Transparency in AI

Consistent with prior research, participants reported mixed trust in AI-driven recommendations, with preferences for tools validated or endorsed by clinicians [7,10]. While generative AI tools were used by a majority of participants, skepticism remained regarding the transparency of underlying data sources and the ability to tailor outputs to individual health contexts. Trust in AI was positively associated with perceived alignment with provider advice and the presence of human oversight, echoing findings from previous trust frameworks in healthcare technology [9,10].

### 6.3. Usability and Health Literacy Gaps

Participants across all segments identified usability challenges, including difficulty interpreting metrics, lack of actionable feedback, and fragmented data across platforms. These findings align with established literature highlighting the importance of user-centered design in mHealth applications [8]. Furthermore, individuals with lower digital or health literacy reported reduced confidence in acting on AI-based insights. These gaps may exacerbate health inequities if not addressed through inclusive design and adaptive educational tools.

### 6.4. Persona-Specific Needs and Design Implications

The segmentation revealed distinct priorities across user personas. Proactive Health Enthusiasts sought advanced features and real-time feedback, while Chronic Condition Managers valued longitudinal tracking and provider integration. Conversely, the Not Tracking group highlighted motivational and cognitive barriers to digital engagement. These insights suggest a need for adaptive interfaces that dynamically tailor content, visualizations, and recommendations based on user context and goals [25].

### 6.5. Implications for Future Digital Health Design

Our findings support the development of human-AI collaboration systems that prioritize explainability, personalization, and trust scaffolding. Incorporating participatory design approaches and feedback loops into digital health development may increase engagement and equity. Furthermore, integration with electronic health records (EHRs) and support for bidirectional communication with care teams were identified as essential features for clinical-grade digital tools.

### 6.6. Limitations

This study is limited by its qualitative nature and sample size, which may not capture the full spectrum of user experiences. Additionally, participants were recruited primarily through digital communities and referrals, potentially biasing the sample toward digitally literate individuals. However, the use of an NSF I-Corps-inspired framework allowed for rapid thematic saturation and identification of actionable insights.

A limitation of this study is that qualitative analysis was conducted by a single analyst without verification procedures such as double-coding or intercoder reliability checks. While analytic memos and saturation monitoring were used to enhance transparency, this approach may have introduced subjectivity. The findings should therefore be interpreted as exploratory and hypothesis-generating, and future work should incorporate multiple coders and triangulation to strengthen rigor.

### 6.7. Methodological Considerations in Qualitative Analysis

This study employed thematic analysis to identify cross-cutting themes from interview data. While thematic analysis is widely used for exploratory qualitative research, our process was limited to a single round of coding conducted by one analyst. No formal double coding, inter-coder reliability statistics, or participant validation (“member checking”) were performed. As a result, the findings should be interpreted as preliminary and hypothesis-generating rather than definitive. Future studies should strengthen methodological rigor by incorporating multiple coders, calculating inter-rater agreement, and using participant validation or triangulation across data sources to enhance trustworthiness. Despite these limitations, the present analysis provides useful initial insights into stakeholder perspectives that informed the framework design.

### 6.8. Model Validity

A key limitation of this work is that the predictive framework has not yet undergone formal validation. Internal validation approaches such as cross-validation, sensitivity testing, or bias analysis were beyond the scope of this preliminary study. Similarly, external validation in independent cohorts will be essential before clinical translation. While the present findings provide conceptual and exploratory evidence, future research must implement rigorous validation protocols to confirm robustness, generalizability, and fairness across populations.

## 7. Conclusions

Digital health technologies must go beyond data collection to deliver meaningful, trusted, and context-aware insights. By integrating usability principles, AI transparency, and personalization strategies tailored to diverse user segments, future systems may enhance both engagement and health outcomes.

This study contributes to the growing body of literature on digital health engagement by presenting a multidimensional, age-stratified analysis of user preferences across device adoption, data sharing, health tracking features, and perceived needs. While previous research has generally examined these dimensions in isolation, our integrated approach offers a composite view of how demographic, behavioral, and psychosocial factors coalesce to inform digital health behavior. By combining structured survey data with visualization and comparative analysis, we provide a roadmap for designing age-responsive and feature-personalized digital health interventions.

From a practical standpoint, our findings emphasize the importance of targeting health interventions not only by clinical condition but also by demographic profile, technological familiarity, and social context. Future work should expand the sample size, incorporate qualitative insights, and examine longitudinal behavior to validate these age-differentiated preferences in real-world digital health adoption scenarios.

## Figures and Tables

**Figure 1 healthcare-13-02753-f001:**
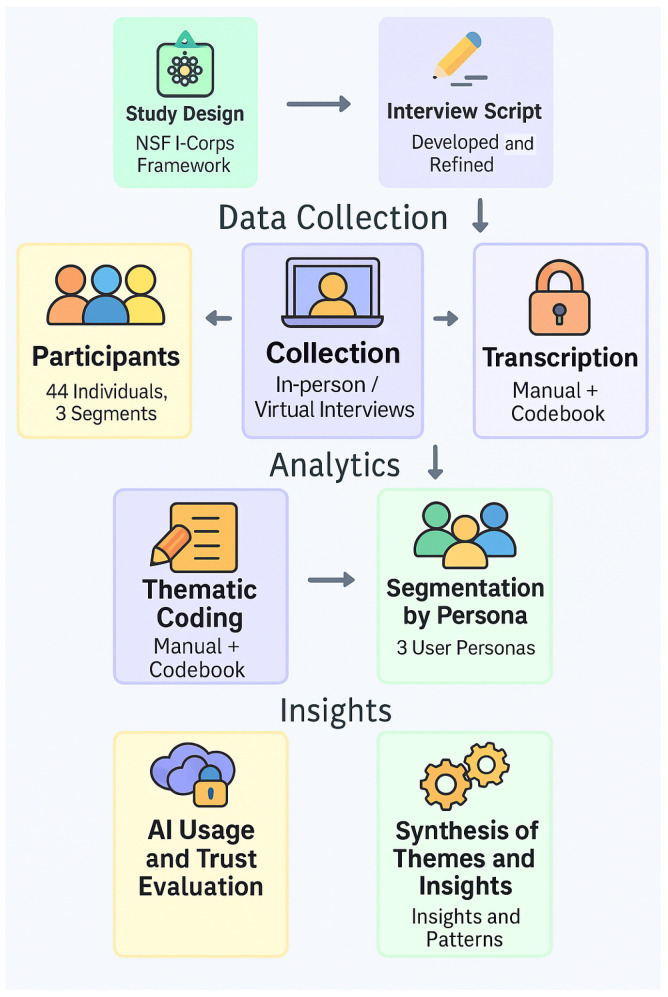
Overview of the study methodology incorporating the NSF I-Corps customer discovery framework.

**Figure 2 healthcare-13-02753-f002:**
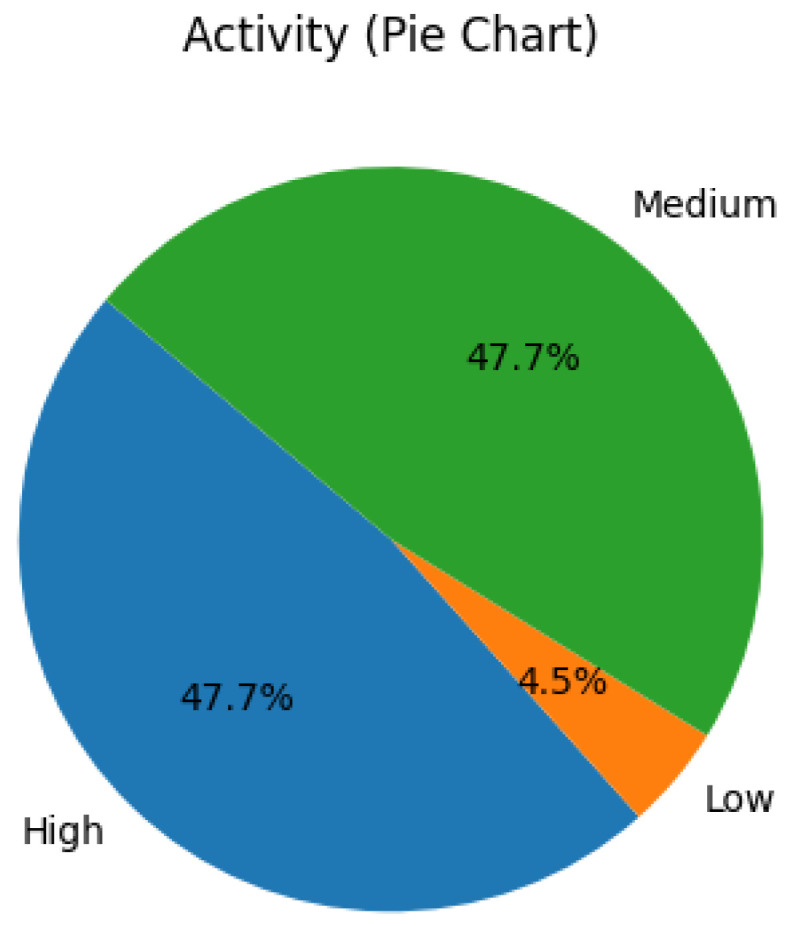
Distribution of participant physical activity levels.

**Figure 3 healthcare-13-02753-f003:**
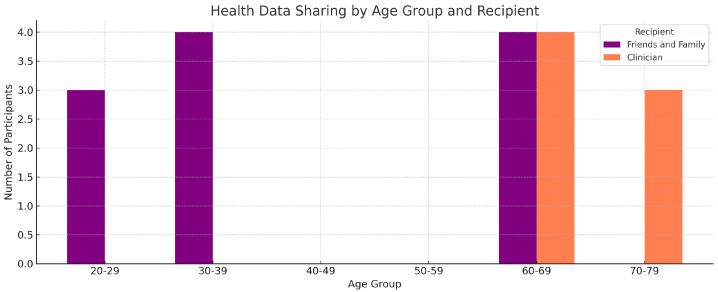
Health data sharing patterns across age groups, segmented by recipient type.

**Figure 4 healthcare-13-02753-f004:**
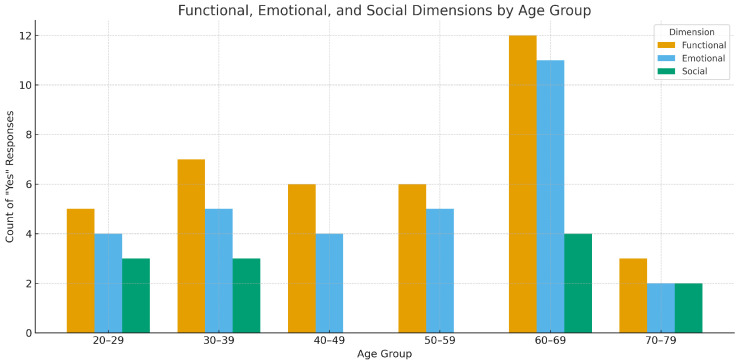
Bar chart showing counts of participants reporting functional, emotional, and social value of digital health tools, stratified by age group.

**Figure 5 healthcare-13-02753-f005:**
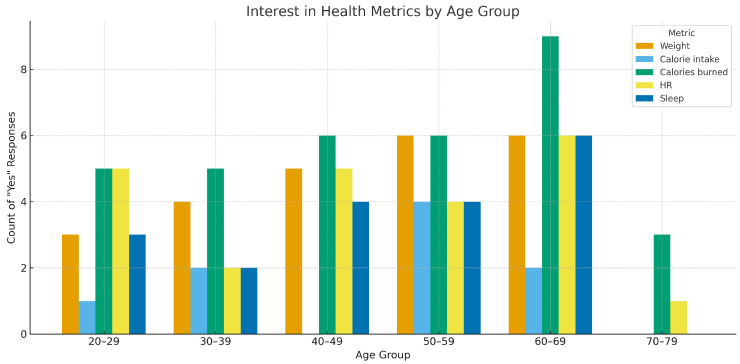
Interest in Health Metrics by Age Group based on affirmative responses.

**Figure 6 healthcare-13-02753-f006:**
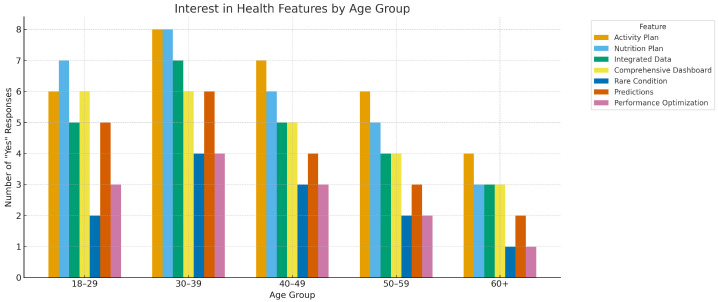
Interest in Digital Health Features by Age Group: Count of “Yes” responses across seven categories.

**Figure 7 healthcare-13-02753-f007:**
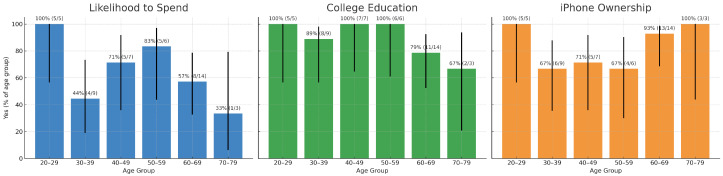
Distribution of affirmative responses by age group for likelihood to spend, college education, and iPhone usage. Error bars indicate 95% Wilson confidence intervals (CI) for binomial proportions, reflecting the uncertainty due to small sample size in each age group.

**Figure 8 healthcare-13-02753-f008:**
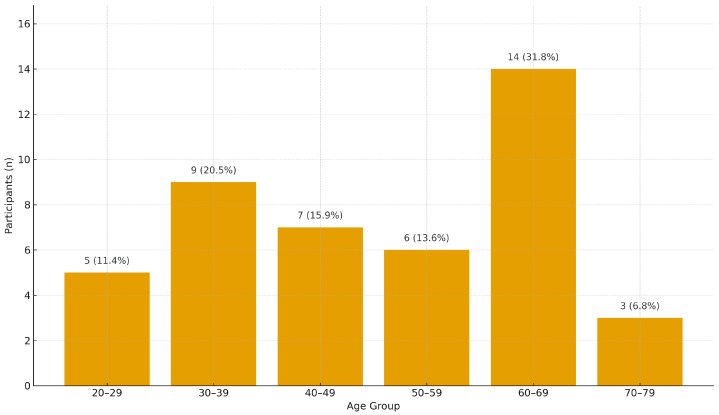
Distribution of ages in the study.

**Figure 9 healthcare-13-02753-f009:**
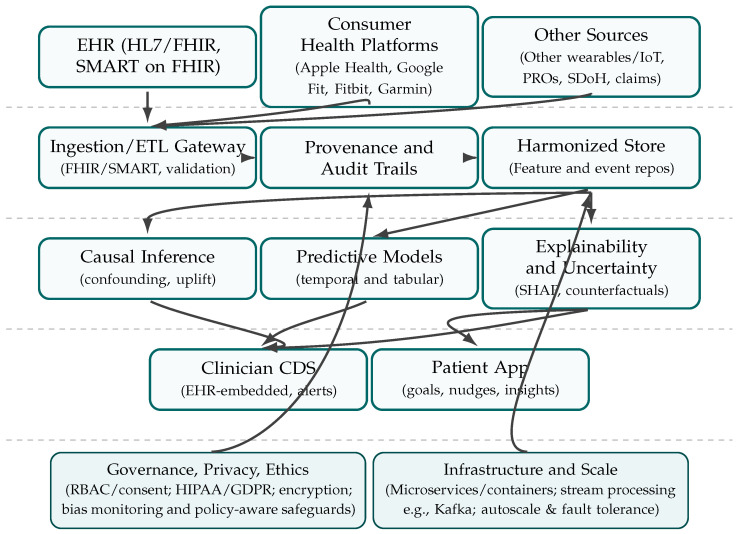
Conceptual Technical framework.

**Table 1 healthcare-13-02753-t001:** Existing work in artificial intelligence in healthcare.

Study/Reference	Focus Area	Key Findings	Implications for This Work
Topol (2019) [1]	High-performance Medicine	AI augments clinical decision-making, enhances diagnostic accuracy	Supports use of AI-driven health monitoring for proactive care
Kvedar & Fogel (2016) [2]	Internet of Healthy Things	Interconnected devices enable real-time patient monitoring	Justifies integration of wearables and EHRs
ONC ISA (2023) [3]	Interoperability Standards	Highlights technical barriers in multimodal data exchange	Reinforces need for seamless EHR and wearable integration
Kvedar & Fogel (2016) [2]	Internet of Healthy Things	Interconnected devices enable real-time patient monitoring	Justifies integration of wearables and EHRs
Yin et al. (2019) [4]	Context-aware Systems	Real-world AI in healthcare often lacks contextual awareness and does not integrate patients’ lived experiences or behavioral data	Highlights the need for emotionally intelligent, context-aware, and personalized health AI
Jiang et al. (2017) [9]	AI in Healthcare	Survey of AI applications in diagnosis and risk modeling	Aligns with predictive modeling focus of this work
Blease et al. (2019) [7]	AI and Patient Trust	Clinician endorsement improves AI acceptance by patients	Justifies explainable AI features and physician–AI collaboration
Glikson & Woolley (2020) [10]	Trust in AI	Human-centered AI interaction design builds trust	Encourages provider-integrated, transparent AI systems
Zhou et al. (2019) [8]	Usability of mHealth Apps	Develops MAUQ to assess mobile health tool usability	Highlights role of user-centered design and testing
Nunes et al. (2015) [25]	Self-care Technologies	Explores tensions in digital self-care adoption	Informs need for behavior-based personalization

**Table 2 healthcare-13-02753-t002:** Summary of existing work in electronic health record (EHR) research.

Study/Reference	Focus Area	Key Findings	Implications for This Work
Weiskopf and Weng (2013) [28]	Data Quality and Completeness	Defined types of missingness in EHRs and proposed metrics like Record Strength Score (RSS)	Provides baseline methods to assess data completeness
Gurupur et al. (2025) [30]	Data Quality and Completeness	Explored ontology-driven modeling of incompleteness in health data systems	Supports design of completeness-aware imputation frameworks
Mandel et al. (2016) [34]	Interoperability	Described SMART on FHIR architecture to standardize cross-system EHR access	Informs API integration for multi-source health data
IEEE P2933 (2021) [27]	Interoperability	Standardized IoT data and device interoperability for clinical use	Relevant for integrating wearable and EHR data pipelines
Chen et al. (2020) [32]	Bias and Health Disparities	Studied effects of incomplete EHRs on marginalized populations	Highlights need for equity-aware data completion methods
Wright et al. (2011) [35]	Clinical Decision Support	Demonstrated how EHR-integrated alerts can improve decision-making	Reinforces value of completeness for accurate alerts
Shortliffe and Sepúlveda (2018) [36]	Clinical Decision Support	Reviewed evolution of EHR-based decision support systems	Emphasizes integration challenges of real-time modeling
Ratwani et al. (2018) [37]	Human–Computer Interaction	Identified usability issues causing clinician fatigue	Underlines role of interface design in adoption of AI tools
Khairat et al. (2018) [38]	Human–Computer Interaction	Assessed EHR usability using objective task scores	Supports need for usability testing in AI-driven systems
Miotto et al. (2016) [39]	Machine Learning	Proposed Deep Patient, a model trained on EHRs to predict disease onset	Demonstrates feasibility of deep learning on raw EHR data
Rajkomar et al. (2018) [40]	Machine Learning	Validated scalable deep learning models for clinical prediction	Affirms predictive power of high-dimensional health data

**Table 3 healthcare-13-02753-t003:** Recent work in wearables and integrated health data.

Study/Reference	Focus Area	Key Findings	Implications for This Work
Piwek et al. (2016) [42]	Adoption of consumer wearables	Widespread adoption of fitness trackers but lack of clinical integration	Highlights user engagement but underscores the need for medical validation and integration
Swan (2012) [47]	Quantified self-movement	Wearables enable self-tracking of diverse health metrics	Supports the growing need for unified health dashboards
Gresham et al. (2020) [48]	Wearables in chronic disease	Demonstrated benefits for diabetes and cardiovascular monitoring	Validates the importance of predictive analytics tools in chronic care
Muoio (2019) [49]	Integration with EHRs (Apple Health)	Data portability from consumer devices into clinical workflows	Shows the feasibility of wearable–EHR integration
Patel et al. (2012) [41]	Wearable biosensors	Review of biosensor tech for personalized health monitoring	Supports multimodal sensor fusion in the proposed tool
Haghi et al. (2017) [50]	Privacy and usability	Explores user trust and data protection in health wearables	Underlines the need for user-centered design and secure interfaces

**Table 4 healthcare-13-02753-t004:** Summary of affirmative responses by age group across health-related dimensions. Notably, participants over the age of 50 showed distinct patterns, including lower reported activity interest and feature interest, but higher reporting of functional and emotional needs. These findings suggest age-related motivational and training considerations.

Category	20–29	30–39	40–49	50–59	60–69	70–79
Likely to Spend	5	4	5	5	8	1
College Educated	5	8	7	6	11	2
iPhone Ownership	5	6	5	4	13	3
Activity Interest	3	4	5	6	6	0
Calories Burned	5	5	6	6	9	3
HR Monitoring	5	2	5	4	6	1
Feature Interest (max)	7–8	8	7	6	4	1–2
Functional Needs	5	7	6	6	12	3
Emotional Needs	4	5	4	5	11	2
Social Needs	3	3	0	0	4	2
Shares with Clinician	0	0	0	0	4	3
Shares with Family	3	4	0	0	4	0

Note: Characteristics of the sample, including qualifications, occupations, and socio-economic status, are documented in the text. Patterns among participants aged 50+ warrant further discussion regarding motivation and the potential role of training interventions.

**Table 5 healthcare-13-02753-t005:** Comparable studies in AI-enabled digital health.

Study	Focus Area	Key Contribution	Relevance to This Study
Topol (2019) [1]	AI in Medicine	Emphasized convergence of human and AI capabilities in clinical care	Supports integration of AI in patient-centered health monitoring
Piwek et al. (2016) [42]	Wearables	Discussed barriers to adoption of consumer wearables	Provides context on usability and engagement issues across demographics
Chen et al. (2020) [32]	AI and Disparities	Explored AI’s potential to reduce healthcare disparities	Informs design of equitable AI health tools
Lupton (2016) [60]	Digital Health Sociology	Analyzed data sharing norms and health self-tracking	Informs age-specific sharing behaviors observed in this study
Miotto et al. (2016) [39]	Deep Learning on EHR Data	Used EHRs from a large hospital system for unsupervised patient representation	Informs predictive modeling using EHRs
Gresham et al. (2020) [48]	Wearables for Monitoring	Evaluated wearable impact on oncology patient outcomes	Corroborates wearable use trends
Nazábal et al. (2020) [21]	Missing Data in Health	Demonstrated VAE-based imputation for heterogeneous data	Supports deep generative approaches to EHR completion
Charness & Boot (2009) [74]	Aging and Tech Use	Highlighted barriers for older adults using digital tools	Supports persona-specific usability findings
This Study	Patient-Centered Predictive Tools	Identified age-stratified needs and trust patterns in AI tools via NSF I-Corps methodology	Extends the literature with design-informed, user-segmented digital health recommendations

## Data Availability

The data supporting the findings of this study consist of de-identified interview transcripts and thematic analysis notes. Due to the sensitive nature of qualitative responses and to protect participant confidentiality, the data are not publicly available. Aggregated summaries or illustrative excerpts may be shared upon reasonable request to the corresponding author, subject to ethical approval.

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
