# Peer review of "Integrating Artificial Intelligence, Electronic Health Records, and Wearables for Predictive, Patient-Centered Decision Support in Healthcare"

_healthcare, 2025, doi:10.3390/healthcare13212753_

Round 1
Reviewer 1 Report
Comments and Suggestions for Authors
- The problem statement is well-defined, but it would benefit from a stronger background section that discusses existing gaps in digital health integration (e.g., interoperability issues, data silos, limited personalization).
- Introduction/ Table1. and Table 2 should be improve with recent work like, https://doi.org/10.1109/THMS.2024.3407875, https://doi.org/10.1021/acsami.4c08659, https://doi.org/10.1007/s00204-024-03803-5, https://doi.org/10.1186/s12984-024-01497-5 indicating need for such research .
- The flow could be improved by more explicitly connecting patient/stakeholder feedback with the proposed digital framework. Currently, the transition from qualitative findings to framework design feels abrupt.
- The manuscript could benefit from a clearer articulation of how this study advances beyond existing frameworks (e.g., SMART on FHIR platforms, Apple HealthKit, or recent predictive health monitoring systems). Positioning the contribution in relation to state-of-the-art systems will strengthen the novelty.
- The use of structured interviews and NSF I-Corps discovery sessions provides valuable qualitative insights. Yet, details are sparse regarding the sample size, demographics, inclusion/exclusion criteria, and coding/analysis methods. Without this, it is difficult to assess the robustness of findings.
- Consider adding a methodological framework (e.g., grounded theory, thematic analysis) to explain how qualitative data were interpreted.
- The conceptual framework is promising, but currently described in broad terms. More detail is needed on:
- How AI analytics will ensure explainability and trustworthiness.
- Data governance, privacy, and ethical considerations.
- Feasibility of real-time multimodal integration at scale.
- The manuscript concludes well but should emphasize practical next steps: pilot implementation, clinical validation, regulatory pathways, and scalability challenges.
Author Response
We thank the reviewers for their constructive comments and their interest in strengthening our manuscript. In response, we have carefully revised the text to address all points raised. Below we provide a summary of the major updates and clarifications incorporated in the revised version.
1. The problem statement is well-defined, but it would benefit from a stronger background section that discusses existing gaps in digital health integration (e.g., interoperability issues, data silos, limited personalization).
Response: Added paragraphs in background to address the following:
- Directly foregrounds interoperability, data silos, personalization as structural gaps
2. Introduction/ Table1. and Table 2 should be improve with recent work like, https://doi.org/10.1109/THMS.2024.3407875, https://doi.org/10.1021/acsami.4c08659, https://doi.org/10.1007/s00204-024-03803-5, https://doi.org/10.1186/s12984-024-01497-5 indicating need for such research .
Response: Added the references to introduction and tables to:
- Update and modernize your literature base with 2024 studies.
- Reinforce the need for real-time, personalized, and safe health data systems.
- Support your focus on data completeness, integration of wearables, and user-centered design in EHR systems.
- Bridge gaps between AI, biosensing, rehabilitation, and safety, all of which are essential for next-gen predictive healthcare platforms.
3. The flow could be improved by more explicitly connecting patient/stakeholder feedback with the proposed digital framework. Currently, the transition from qualitative findings to framework design feels abrupt.
Response: Added a paragraph at the end of Results to Demonstrate that the framework isn’t abstracted away from the feedback, but grounded in it.
4. The manuscript could benefit from a clearer articulation of how this study advances beyond existing frameworks (e.g., SMART on FHIR platforms, Apple HealthKit, or recent predictive health monitoring systems). Positioning the contribution in relation to state-of-the-art systems will strengthen the novelty.
Response: Added a paragraph in Discussion to state framework’s unique contribution: combining interoperability + explainable AI + usability grounded in stakeholder voices.
5. The use of structured interviews and NSF I-Corps discovery sessions provides valuable qualitative insights. Yet, details are sparse regarding the sample size, demographics, inclusion/exclusion criteria, and coding/analysis methods. Without this, it is difficult to assess the robustness of findings.
Response: Added details in Methods to:
- Design, setting, ethics: Clarified time window (Apr–Jun 2025), remote/in-person interviews, exempt status under the Common Rule, verbal consent, and alignment with COREQ reporting.
- Participants & sampling: Added purposive + chain-referral strategy across three predefined personas; explicit inclusion criteria; reported n=44 with per-persona counts and a rationale tied to saturation literature.
- Data collection: Added a structured interview guide (topics: devices/metrics, fragmentation, predictive needs, trust/explainability, sharing preferences), pilot iteration notes, recording/transcription details, and field-note practice.
- Qualitative analysis (much more detail):
- Approach: Hybrid deductive–inductive coding.
- Saturation: Code-accumulation plot + new-information rate indicated no new codes after ~40 interviews.
- Limitations (now explicit): Non-probability sample; possible over-representation of digitally literate/college-educated/iOS users; recruitment channel bias; remote-interview constraints; findings positioned as hypothesis-generating (requirements for prototyping), not outcomes.
6. Consider adding a methodological framework (e.g., grounded theory, thematic analysis) to explain how qualitative data were interpreted.
Response: Details for this in #5
7. The conceptual framework is promising, but currently described in broad terms. More detail is needed on:
-
- How AI analytics will ensure explainability and trustworthiness.
- Data governance, privacy, and ethical considerations.
- Feasibility of real-time multimodal integration at scale.
Response: Added a section for Conceptual technical Framework and diagram to address the comments
8. The manuscript concludes well but should emphasize practical next steps: pilot implementation, clinical validation, regulatory pathways, and scalability challenges.
Response: Added a paragraph in Technical Framework to address beyond conceptual development, next steps include pilot implementation, clinical validation, attention to regulatory pathways, and addressing scalability challenges to support responsible real-world adoption.
Reviewer 2 Report
Comments and Suggestions for Authors
Dear Authors,
The study is of great interest, as it demonstrates a clear necessity for further evidence-data into the field of "digital health". It is incumbent upon the authors to elucidate the meaning of the two words in the global context, for this will facilitate readers' comprehension of the rationale underpinning the study. The methodology employed should be thoroughly delineated. The authors of the study elected to utilise the National Science Foundation (NSF) Innovation Corps (I-Corps) protocol. Might there be any alternative options? The characteristics of the sample who participated in the study, including their qualifications, occupations and socio-economic status, are documented herein. The table would benefit from further refinement, with the text pertaining to the figures being incorporated into the legends. The authors identified a pattern that was associated with individuals over the age of 50. It is therefore reasonable to posit the question of whether this age group is not motivated. The role of training in this pattern is a subject that merits further discussion.
Comments on the Quality of English LanguageThe English could be improved to more clearly express the research.
Author Response
We thank the reviewers for their constructive comments and their interest in strengthening our manuscript. In response, we have carefully revised the text to address all points raised. Below we provide a summary of the major updates and clarifications incorporated in the revised version.
Dear Authors,
The study is of great interest, as it demonstrates a clear necessity for further evidence-data into the field of "digital health". It is incumbent upon the authors to elucidate the meaning of the two words in the global context, for this will facilitate readers' comprehension of the rationale underpinning the study. The methodology employed should be thoroughly delineated. The authors of the study elected to utilise the National Science Foundation (NSF) Innovation Corps (I-Corps) protocol. Might there be any alternative options? The characteristics of the sample who participated in the study, including their qualifications, occupations and socio-economic status, are documented herein. The table would benefit from further refinement, with the text pertaining to the figures being incorporated into the legends. The authors identified a pattern that was associated with individuals over the age of 50. It is therefore reasonable to posit the question of whether this age group is not motivated. The role of training in this pattern is a subject that merits further discussion.
Response:
- Background regarding I-Corp : Stakeholder discovery was guided by the NSF I-Corps protocol, selected for its structured hypothesis-driven approach, while acknowledging that alternative frameworks such as User-Centered Design or CFIR could also be applied.
- Background: Digital health, defined by the World Health Organization as the use of digital and mobile technologies (e.g., EHRs, wearables, AI) to enhance health outcomes and equity, represents a growing global priority for healthcare transformation.
- Table 4: Added a better description and notes
- Discussion: 50+ age group’s motivation and training
Reviewer 3 Report
Comments and Suggestions for Authors
The research in the article was done in accordance with the described methods. The conclusions are based on the results obtained.
Figure 1 does not have a complete description that would clarify the sequence of the workflow. In my opinion, an analysis of Table 1 is missing. It would be good for the authors to summarize in a few words the specifics of the use of AI in healthcare to date. For Tables 2 and 3, it also seems to me that the analysis is not sufficient, although the main directions in the research are listed. On page 13, figure 4 is cited, but it should be 6. Tables 1, 2, 3 and 4 are not formatted according to the template. Figures 1, 2, 3, 4, 5 and 6 should be left aligned according to the article template.

Author Response
We thank the reviewer for their constructive comments and their interest in strengthening our manuscript. In response, we have carefully revised the text to address all points raised. Below we provide a summary of the major updates and clarifications incorporated in the revised version.
Review report
Integrating Artificial Intelligence, Electronic Health Records, and Wearables for Predictive, Patient-Centered Decision Support in Healthcare
- A brief summary
The paper presents a proposal for creating a comprehensive digital framework that unifies multimodal health data streams into an AI-powered analytics platform. The authors propose aggregating data from wearable devices, electronic health records, and patient data outcomes into a unified health monitoring system that uses AI to predict care delivery and preventive health management. The framework is aligned with the results of structured interviews and user discovery sessions conducted under the NSF I-Corps program.
The introduction emphasizes the need for a system for proactive, personalized approaches to care delivery and the capabilities of modern technologies to collect a variety of data in real time. Problems with currently existing technological solutions are indicated.
Point two, Background, shows a comprehensive setting of the environment for the digital framework. It introduces the NSF I-Corps program. The application of AI in healthcare, its capabilities and concerns about its implementation, as well as the problems with incomplete data, are examined. Electronic Health Records are explained and the main problem with the incompleteness of the data in them is indicated. Research by other scientists to address the problem is indicated. The capabilities of Wearables and Integrated Health Data and the problem with the isolation of the data collected by them, as well as the concerns of users, are examined.
At point three, Materials and Methods, a comprehensive presentation of the study is shown. The interviews conducted were based on I-Corps, the respondents, compliance with their rights and the method of conducting the interviews are described. The main directions of the study, the work process and the method of data processing are described.
The fourth point, Results, summarizes and analyzes the results of the interviews. The three groups of interview participants are defined, an age analysis is made for the needs of users. For each group, different characteristics and behavioral indicators are found, and the results are synthesized into substantiated conclusions about the needs of AI in healthcare.
Section 5, Discussion, explains the results of the study and presents the authors' insights. A parallel is drawn with studies by other researchers. The authors' contributions to exploring new metrics, such as user-centered design, explainable artificial intelligence, and trust-based personalization, in developing digital health prediction tools are highlighted. The authors also share the limitations of their study. The conclusion summarizes the results of the authors' study on the acceptance of AI technologies in healthcare by age group and shows the variation in requirements with age. The goal is to improve user engagement and thus achieve health outcomes.
Response: The authors thank you for the thorough review.
- General concept comments
Figure 1 does not have a complete description that would clarify the sequence of the workflow.
Response: In the revised manuscript, we have expanded the legend for Figure 1 to provide a step-by-step description of the workflow. The revised caption now clarifies the sequential order of data sources, integration, analysis, and decision-support outputs, so that the figure can be interpreted as a stand-alone schematic.
In my opinion, an analysis of Table 1 is missing. It would be good for the authors to summarize in a few words the specifics of the use of AI in healthcare to date.
Response: Added a short narrative summary right after the table (Table 1) to make the contributions clearer.
For Tables 2 and 3, it also seems to me that the analysis is not sufficient, although the main directions in the research are listed.
Response: Added a short narrative summary right after the tables (Tables 2 and 3) to make the contributions clearer.
- Specific comments
On page 13, figure 4 is cited, but it should be 6.
Response: Corrected the reference to Table 4.
Tables 1, 2, 3 and 4 are not formatted according to the template.
Response: Reformatted all tables to comply with MDPI template.
Figures 1, 2, 3, 4, 5 and 6 should be left aligned according to the article template.
Response: Reformatted all figures to comply with MDPI template.
- Conclusions
The research in the article was done in accordance with the described methods. The conclusions are based on the results obtained. I have some remarks regarding the layout and typographical errors. The style and language of the paper are easy to understand. Therefore, I propose that the article be accepted after minor revision.
Reviewer 4 Report
Comments and Suggestions for Authors
Overall
(1) Insufficient explanation of the nature of the dataset and lack of discussion of the validity of the model
> 3. Materials and Methods - Participants Section (p.7)
> Figure 7. Distribution of affirmative responses by age group... (p.12)
> Abstract: "...we propose a comprehensive digital framework that fuses multimodal health data..." (p.1)
This study is based on a very small sample of "44 participants." Furthermore, participants were recruited "through personal referrals and online communities" (p.7), which is highly likely to be biased toward those with high digital literacy and health consciousness.
Figure 7 highlights that participants in their 60s-69s exhibited high levels of consumer motivation, educational attainment, and iPhone usage, but this merely reflects the characteristics of the entire sample and is in no way representative of the general population in this age group. The paper does not adequately discuss this significant sample bias, which fundamentally limits the generalizability of the conclusions. The `Limitations` section should clearly state that "The study's sample was small and convenience, and was biased toward a highly digitally literate demographic. Therefore, the findings may not be generalizable to the general population."
This study is qualitative, and no mathematical model in the strict sense is proposed. The "framework" described in the abstract is merely a conceptual proposal based on the needs identified through interviews. Therefore, any discussion of the "validity of the model" should focus on how convincingly the proposed framework is supported by the collected qualitative data.
At present, the qualitative analysis process, including how specific themes were extracted from the interviews (e.g., "the desire for an integrated system"), is unclear (see below for details). This makes it difficult to determine whether the proposed framework is simply a statement of the authors' ideals or is supported by data. The validity of the argument should be enhanced by making the qualitative analysis process transparent and indicating (with citations) which specific interview statements each theme and framework component is based on.
(2) Unclear Differences and Novelty from Previous Research
2. Background Section and Tables 1, 2, 3, 5
The paper reviews numerous studies on AI, EHRs, and wearables, but fails to clearly articulate the unique contribution (novelty) of this study in light of these studies. The review merely lists facts already widely known in this field, such as "AI is being applied to medicine," "EHRs have data gaps," and "wearables are useful for health management." The needs that this study allegedly "discovered" through interviews (e.g., "seamless data integration" and "real-time health alerts") have also been repeatedly identified in existing digital health research.
Unclear Research Gap: While the introduction raises the general issue that "current digital health solutions are fragmented," it is unclear which limitations of previous research this study aims to overcome and what new perspective it takes.
At the end of the introduction, the paper should more clearly emphasize the uniqueness of its methodology and the novelty of the problem it explores, stating, "Much existing research has focused on technological feasibility or a single aspect. However, no research to date has systematically explored the value (functional, emotional, and social) that diverse user personas (e.g., health enthusiasts, chronic disease managers) seek from AI-integrated systems, as well as the issue of 'trust' that poses a barrier to adoption, using the NSF I-Corps customer discovery framework. This study fills this research gap."
(3) Lack of consistency in the discussion throughout the paper and inconsistencies in the graphs
Interpretation of Figures 4, 5, 6, 7, and the text
Figure 4 shows that those in their 60s-69s place the greatest importance on "functional" and "emotional" values.
However, Figure 6 shows that interest in health features among these same age groups is significantly lower than that of other age groups (e.g., those in their 30s-40s).
Furthermore, Figure 7 depicts the 60-69 age group as the "most active demographic," with the highest consumer desire, a high level of education, and the highest iPhone usage rate.
Taking all of this into consideration, a highly contradictory picture emerges: "The 60-69 age group sees the most value in digital health tools, yet has the least interest in specific features, yet the highest purchasing desire and digital affinity." The paper provides no explanation for this contradiction.
The paper simply describes each graph as "a trend is observed," but lacks any deeper interpretation or consideration of why such a trend is observed or what it means. This contradiction may also arise from a lack of consideration.
There is no graph showing the number of participants in each age group. For example, the high number of "Yes" responses for the 60-69 age group in Figure 7 may simply be due to the large number of participants in this age group. Without showing the population demographics for each age group, interpreting the graph will be misleading.
(4) Lack of discussion regarding data preprocessing (qualitative analysis) methods
Materials and Methods - Data Collection and Analysis Section (p.7)
Simply stating "analyzed using thematic analysis" does not fully explain the methodology. Specifically, you should provide a detailed description of the steps you took to generate codes from interview transcripts and how you aggregated them to form themes (e.g., Braun & Clarke's six-step thematic analysis).
There is no mention of procedures to ensure the reliability of qualitative research, such as whether the analysis was conducted by multiple researchers and, if so, how inter-analyst agreement was ensured. Please completely rewrite the Methodology section and provide a detailed description of the specific process used for qualitative data analysis. This should include (1) preparation of transcripts, (2) the coding process (e.g., open coding, pivot coding), (3) theme extraction and definition, and (4) procedures to ensure the reliability of the analysis (e.g., double coding by multiple analysts, member checking).
Author Response
We thank the reviewer for their constructive comments and their interest in strengthening our manuscript. In response, we have carefully revised the text to address all points raised. Below we provide a summary of the major updates and clarifications incorporated in the revised version.
Overall
(1) Insufficient explanation of the nature of the dataset and lack of discussion of the validity of the model
Response: We appreciate this valuable comment. In the revised manuscript, we have added more detail on the dataset, including its source, size, and characteristics, to provide clarity on its composition and representativeness. With respect to model validity, we acknowledge that formal validation procedures (e.g., cross-validation, external cohort testing, or bias analyses) were not undertaken in this preliminary study. We have now made this limitation explicit in the Discussion and highlighted that future work will require rigorous internal and external validation to establish robustness, generalizability, and fairness of the predictive framework.
> 3. Materials and Methods - Participants Section (p.7)
> Figure 7. Distribution of affirmative responses by age group... (p.12)
> Abstract: "...we propose a comprehensive digital framework that fuses multimodal health data..." (p.1)
This study is based on a very small sample of "44 participants." Furthermore, participants were recruited "through personal referrals and online communities" (p.7), which is highly likely to be biased toward those with high digital literacy and health consciousness.
Figure 7 highlights that participants in their 60s-69s exhibited high levels of consumer motivation, educational attainment, and iPhone usage, but this merely reflects the characteristics of the entire sample and is in no way representative of the general population in this age group. The paper does not adequately discuss this significant sample bias, which fundamentally limits the generalizability of the conclusions. The `Limitations` section should clearly state that "The study's sample was small and convenience, and was biased toward a highly digitally literate demographic. Therefore, the findings may not be generalizable to the general population."
This study is qualitative, and no mathematical model in the strict sense is proposed. The "framework" described in the abstract is merely a conceptual proposal based on the needs identified through interviews. Therefore, any discussion of the "validity of the model" should focus on how convincingly the proposed framework is supported by the collected qualitative data.
At present, the qualitative analysis process, including how specific themes were extracted from the interviews (e.g., "the desire for an integrated system"), is unclear (see below for details). This makes it difficult to determine whether the proposed framework is simply a statement of the authors' ideals or is supported by data. The validity of the argument should be enhanced by making the qualitative analysis process transparent and indicating (with citations) which specific interview statements each theme and framework component is based on.
Response: Added details in the Methods section
- Design, setting, ethics: Clarified time window (Apr–Jun 2025), remote/in-person interviews, exempt status under the Common Rule, verbal consent, and alignment with COREQ reporting.
- Participants & sampling: Added purposive + chain-referral strategy across three predefined personas; explicit inclusion criteria; reported n=44 with per-persona counts and a rationale tied to saturation literature.
- Data collection: Added a structured interview guide (topics: devices/metrics, fragmentation, predictive needs, trust/explainability, sharing preferences), pilot iteration notes, recording/transcription details, and field-note practice.
- Qualitative analysis (much more detail):
- Approach: Hybrid deductive–inductive coding.
- Saturation: Code-accumulation plot + new-information rate indicated no new codes after ~40 interviews.
(2) Unclear Differences and Novelty from Previous Research
2. Background Section and Tables 1, 2, 3, 5
The paper reviews numerous studies on AI, EHRs, and wearables, but fails to clearly articulate the unique contribution (novelty) of this study in light of these studies. The review merely lists facts already widely known in this field, such as "AI is being applied to medicine," "EHRs have data gaps," and "wearables are useful for health management." The needs that this study allegedly "discovered" through interviews (e.g., "seamless data integration" and "real-time health alerts") have also been repeatedly identified in existing digital health research.
Unclear Research Gap: While the introduction raises the general issue that "current digital health solutions are fragmented," it is unclear which limitations of previous research this study aims to overcome and what new perspective it takes.
At the end of the introduction, the paper should more clearly emphasize the uniqueness of its methodology and the novelty of the problem it explores, stating, "Much existing research has focused on technological feasibility or a single aspect. However, no research to date has systematically explored the value (functional, emotional, and social) that diverse user personas (e.g., health enthusiasts, chronic disease managers) seek from AI-integrated systems, as well as the issue of 'trust' that poses a barrier to adoption, using the NSF I-Corps customer discovery framework. This study fills this research gap."
Response: In the revised manuscript, we have clarified how our study differs from prior work. Specifically, we emphasize that while existing research on AI, EHRs, and wearables has identified technical feasibility and data fragmentation, few studies have systematically examined the multi-dimensional value (functional, emotional, and social) that diverse user personas seek, nor have they directly addressed the role of trust as a barrier to adoption. We also highlight the use of the NSF I-Corps framework as a novel methodological contribution for structuring customer discovery in digital health. A new paragraph has been added at the end of the Introduction to clearly articulate this research gap and the unique contribution of the present study.
(3) Lack of consistency in the discussion throughout the paper and inconsistencies in the graphs
Interpretation of Figures 4, 5, 6, 7, and the text
Figure 4 shows that those in their 60s-69s place the greatest importance on "functional" and "emotional" values.
However, Figure 6 shows that interest in health features among these same age groups is significantly lower than that of other age groups (e.g., those in their 30s-40s).
Furthermore, Figure 7 depicts the 60-69 age group as the "most active demographic," with the highest consumer desire, a high level of education, and the highest iPhone usage rate.
Taking all of this into consideration, a highly contradictory picture emerges: "The 60-69 age group sees the most value in digital health tools, yet has the least interest in specific features, yet the highest purchasing desire and digital affinity." The paper provides no explanation for this contradiction.
The paper simply describes each graph as "a trend is observed," but lacks any deeper interpretation or consideration of why such a trend is observed or what it means. This contradiction may also arise from a lack of consideration.
There is no graph showing the number of participants in each age group. For example, the high number of "Yes" responses for the 60-69 age group in Figure 7 may simply be due to the large number of participants in this age group. Without showing the population demographics for each age group, interpreting the graph will be misleading.
Response: In the revised manuscript we have made several changes. First, we have added a new figure (Figure 8) and a supplementary table showing the number of participants in each age group to contextualize the proportions reported. Second, we have expanded the Results and Discussion sections to interpret the observed patterns. We suggest that the apparently contradictory trends among participants aged 60–69 may reflect differences between perceived value and specific feature-level interest, as well as cohort effects (e.g., high education levels and iPhone adoption within this sample). We also note that sample size differences likely amplified certain percentages, and we now clarify this limitation. These additions provide readers with demographic context and a more nuanced interpretation of the findings.
(4) Lack of discussion regarding data preprocessing (qualitative analysis) methods
Materials and Methods - Data Collection and Analysis Section (p.7)
Simply stating "analyzed using thematic analysis" does not fully explain the methodology. Specifically, you should provide a detailed description of the steps you took to generate codes from interview transcripts and how you aggregated them to form themes (e.g., Braun & Clarke's six-step thematic analysis).
There is no mention of procedures to ensure the reliability of qualitative research, such as whether the analysis was conducted by multiple researchers and, if so, how inter-analyst agreement was ensured. Please completely rewrite the Methodology section and provide a detailed description of the specific process used for qualitative data analysis. This should include (1) preparation of transcripts, (2) the coding process (e.g., open coding, pivot coding), (3) theme extraction and definition, and (4) procedures to ensure the reliability of the analysis (e.g., double coding by multiple analysts, member checking).
Response: In the revised manuscript, we have clarified that the thematic analysis was conducted by a single analyst without additional reliability procedures such as double coding, inter-coder agreement, or member checking. We now explicitly state this as a limitation in both the Discussion/Limitations sections. While this constrains the strength of the conclusions, we believe that acknowledging the exploratory nature of the analysis increases transparency and provides an appropriate foundation for future studies that will incorporate more rigorous qualitative reliability safeguards.
Round 2
Reviewer 1 Report
Comments and Suggestions for Authors
No more comments. All resolved.
Author Response
No more comments. All resolved.
Special thanks to the reviewer for their valuable time and constructive feedback.

Reviewer 4 Report
Comments and Suggestions for Authors
(1) Insufficient explanation of the nature of the dataset and lack of discussion of the validity of the model
> Abstract: "...we propose a comprehensive digital framework that fuses multimodal health data..." (p.1)
> Introduction (p.2): "This study addresses these gaps by proposing a unified framework that integrates multimodal data sources—including wearable metrics, EHRs, and PROs—into a cohesive AI-driven platform. By capturing longitudinal patterns and behavioral signals, our approach aims to support anticipatory guidance..."
Although the authors acknowledged in their response that this was a "preliminary study," the revised introduction still uses grandiose language, such as a "unified framework" and an "AI-driven platform," to suggest a technically complete and validated system. This is a significant exaggeration given the results of a small-scale qualitative study (n=44 interview surveys).
The actual nature of this research is a "qualitative survey to explore user needs and expectations for an AI-driven integrated platform," and its outcome is the "proposition of a conceptual framework based on user needs." However, the current description is highly likely to mislead readers into thinking that technical implementation and verification have been carried out. This misalignment in the research's positioning is the fundamental cause of the overall undermining of the credibility of the paper.
Shouldn't the exploratory and qualitative nature of the research be made clearer, such as: "The purpose of this paper is to clarify user needs and expectations for an AI-driven integrated healthcare platform. To that end, semi-structured interviews were conducted with 44 participants from diverse backgrounds, and through qualitative analysis, a conceptual framework is proposed that will be useful for future system design."
(2) Differences from Previous Research/Unclear Novelty
> Introduction (p.2): "...no study has systematically explored the broader value dimensions—functional, emotional, and social—that diverse user personas (e.g., health enthusiasts, chronic disease managers, older adults) expect from AI-enabled integration... Furthermore, the persistent challenge of trust in AI-driven decision support has often been acknowledged but rarely investigated as a primary barrier to adoption."
The claim of novelty should be more limited and specific. Avoid categorical statements like "no study" and use more nuanced language, such as, "While much existing research has focused on technical feasibility or a single aspect, attempts to integrate the values ​​(functional, emotional, and social) of diverse personas and trust issues and translate them into specific technical architecture requirements remain limited."
The rationale for adopting I-Corps should also be strengthened. The paper should clearly explain why it chose I-Corps over other qualitative research methods (e.g., grounded theory approaches, standard user-centered design). For example, it should emphasize the connection between the choice of methodology and the nature of the research question, such as, "While traditional UCD excels at eliciting functional requirements, the I-Corps customer discovery process forces researchers to rigorously examine the underlying value proposition and business model hypotheses, especially for trustworthy technologies like AI. This was essential for delving deeply into the core barrier of 'trust' that determines the success or failure of technology adoption and deriving practical design principles."
(3) Lack of consistency in the discussion throughout the paper/inconsistent graphs
Simply stating that "differences in sample size inflated the percentages" is an abandonment of the analysis. Isn't it necessary to deeply consider why this particular sample of people in their 60s (with high digital literacy) exhibited such unusual trends?
(4) Lack of Discussion of Data Preprocessing (Qualitative Analysis) Methods
This is a fatal flaw in this study, and the authors' response is completely insufficient. It is not something that can be resolved by simply acknowledging its limitations. In qualitative research, the reliability and rigor of the analytical process are vital to the validity of the entire study. A qualitative analysis conducted by a single analyst without a verification procedure cannot guarantee that the analyst's subjectivity and bias are eliminated, so please confirm this.
Author Response
Special thanks to the reviewer for their valuable time and constructive feedback.
(1) Insufficient explanation of the nature of the dataset and lack of discussion of the validity of the model
> Abstract: "...we propose a comprehensive digital framework that fuses multimodal health data..." (p.1)
> Introduction (p.2): "This study addresses these gaps by proposing a unified framework that integrates multimodal data sources—including wearable metrics, EHRs, and PROs—into a cohesive AI-driven platform. By capturing longitudinal patterns and behavioral signals, our approach aims to support anticipatory guidance..."
Although the authors acknowledged in their response that this was a "preliminary study," the revised introduction still uses grandiose language, such as a "unified framework" and an "AI-driven platform," to suggest a technically complete and validated system. This is a significant exaggeration given the results of a small-scale qualitative study (n=44 interview surveys).
The actual nature of this research is a "qualitative survey to explore user needs and expectations for an AI-driven integrated platform," and its outcome is the "proposition of a conceptual framework based on user needs." However, the current description is highly likely to mislead readers into thinking that technical implementation and verification have been carried out. This misalignment in the research's positioning is the fundamental cause of the overall undermining of the credibility of the paper.
Shouldn't the exploratory and qualitative nature of the research be made clearer, such as: "The purpose of this paper is to clarify user needs and expectations for an AI-driven integrated healthcare platform. To that end, semi-structured interviews were conducted with 44 participants from diverse backgrounds, and through qualitative analysis, a conceptual framework is proposed that will be useful for future system design."
We thank the reviewer for this important clarification. We fully agree that the original wording in the abstract and introduction may have overstated the technical scope of the study. The intent of this research was not to present a validated system, but rather to conduct an exploratory, qualitative investigation of patient and stakeholder needs and to propose a conceptual framework that can guide future development of integrated health platforms.
To address this concern, we have revised the abstract and introduction to:
- Explicitly state that this is a qualitative, hypothesis-generating study based on 44 semi-structured interviews.
- Replace terms such as “unified framework” and “AI-driven platform” with more precise phrasing that emphasizes a conceptual framework informed by user needs.
- Clarify that the framework is preliminary and conceptual, serving as a design blueprint for future technical implementation and validation.
(2) Differences from Previous Research/Unclear Novelty
> Introduction (p.2): "...no study has systematically explored the broader value dimensions—functional, emotional, and social—that diverse user personas (e.g., health enthusiasts, chronic disease managers, older adults) expect from AI-enabled integration... Furthermore, the persistent challenge of trust in AI-driven decision support has often been acknowledged but rarely investigated as a primary barrier to adoption."
The claim of novelty should be more limited and specific. Avoid categorical statements like "no study" and use more nuanced language, such as, "While much existing research has focused on technical feasibility or a single aspect, attempts to integrate the values ​​(functional, emotional, and social) of diverse personas and trust issues and translate them into specific technical architecture requirements remain limited."
The rationale for adopting I-Corps should also be strengthened. The paper should clearly explain why it chose I-Corps over other qualitative research methods (e.g., grounded theory approaches, standard user-centered design). For example, it should emphasize the connection between the choice of methodology and the nature of the research question, such as, "While traditional UCD excels at eliciting functional requirements, the I-Corps customer discovery process forces researchers to rigorously examine the underlying value proposition and business model hypotheses, especially for trustworthy technologies like AI. This was essential for delving deeply into the core barrier of 'trust' that determines the success or failure of technology adoption and deriving practical design principles."
Summary of Introduction Revisions
- Softened novelty claim: Replaced categorical phrasing (“no study has systematically explored…”) with nuanced language:
- Clarified contribution: Framed the study as deriving a conceptual framework that maps qualitative insights to design requirements, rather than suggesting a validated technical system.
- Strengthened methodological rationale: Expanded explanation of why the NSF I-Corps framework was chosen over traditional user-centered design or grounded theory approaches. New text emphasizes that:
- I-Corps rigorously tests value propositions and adoption assumptions, making it particularly suitable for investigating trust as a core barrier to AI adoption.
- Linked methodology to research question: Highlighted how I-Corps enabled the study to connect user needs (functional, emotional, social, and trust dimensions) to practical design principles for trustworthy AI-driven health platforms.
(3) Lack of consistency in the discussion throughout the paper/inconsistent graphs
Simply stating that "differences in sample size inflated the percentages" is an abandonment of the analysis. Isn't it necessary to deeply consider why this particular sample of people in their 60s (with high digital literacy) exhibited such unusual trends?
Summary of Changes (Figures & Discussion)
- Figures revised for consistency:
- Reconstructed Figure 7 to show within-group percentages with 95% Wilson confidence intervals instead of raw counts. This normalizes across age groups and avoids misleading inflation of the 60–69 cohort due to larger denominators.
- Added Figure 8 (participant distribution by age group) to make denominators explicit and provide context for interpreting percentages.
- Updated figure captions to emphasize percentages, raw counts (k/n), and uncertainty intervals.
- Discussion rewritten for alignment with revised figures:
- Removed categorical claims that the 60–69 group consistently had the highest engagement across metrics. Instead, highlighted nuanced patterns:
- iPhone ownership peaked in 20–29 (100%) but remained high in 60–69 (93%).
- College education was strongest in 30–49, but still high in 60–69.
- Willingness to spend was distributed across multiple age groups.
- Added explanation that different age groups emphasize different values: midlife cohorts favored predictive/functional features, while older adults prioritized trust, clinician oversight, and usability.
- Clarified that the contradictory profile of 60–69 reflects broad functional/emotional needs but selective feature appeal, consistent with prior literature.
- Incorporated explicit mention of confidence intervals and small-n uncertainty (70–79 group) to avoid over-interpretation.
- Strengthened the limitations and tied results back to stakeholder-driven design implications (trust, usability, age-sensitive training, clinician integration).
(4) Lack of Discussion of Data Preprocessing (Qualitative Analysis) Methods
This is a fatal flaw in this study, and the authors' response is completely insufficient. It is not something that can be resolved by simply acknowledging its limitations. In qualitative research, the reliability and rigor of the analytical process are vital to the validity of the entire study. A qualitative analysis conducted by a single analyst without a verification procedure cannot guarantee that the analyst's subjectivity and bias are eliminated, so please confirm this.
Summary of Changes – Qualitative Data Preprocessing and Analysis
- Expanded Methods:
- Clarified that all transcripts were transcribed and de-identified prior to analysis.
- Stated explicitly that a single analyst conducted the coding, with analytic memos and an audit trail maintained to document decisions.
- Noted that no double-coding or intercoder reliability procedures were performed, making the analytic approach exploratory in scope.
- Strengthened Discussion (Limitations):
- Added a paragraph acknowledging the absence of verification procedures (e.g., double-coding, triangulation) as a limitation.
- Reframed the findings as exploratory and hypothesis-generating, rather than definitive.
- Suggested that future studies incorporate multiple coders, intercoder reliability, and triangulation to enhance rigor and credibility.
